

# TIM: Modelling pathways to meet Ireland's long-term energy system challenges with the TIMES-Ireland Model (v1.0)

Olexandr Balyk[1,2], James Glynn[1,2,3], Vahid Aryanpur[1,2], Ankita Gaur[1,2], Jason McGuire[1,2], Andrew Smith[1,2], Xiufeng Yue[1,2,4], and Hannah Daly[1,2]

[1]Energy Policy and Modelling Group, SFI MaREI Centre for Energy, Climate and Marine, Environmental Research Institute, University College Cork, Ireland
[2]School of Engineering, University College Cork, Ireland
[3]Center on Global Energy Policy, School of International and Public Affairs, Columbia University, USA
[4]Dalian University of Technology, China

**Correspondence:** Olexandr Balyk (olexandr.balyk@ucc.ie) or Hannah Daly (h.daly@ucc.ie)

**Abstract.** Ireland has significantly increased its climate mitigation ambition, with a recent government commitment to reduce greenhouse-gases by an average of 7% per year in the period to 2030 and a "net-zero" target for 2050, underpinned by a series of five-year carbon budgets. Energy systems optimisation modelling (ESOM) is a widely-used tool to inform pathways to address long-term energy challenges. This article describes a new ESOM developed to inform Ireland's energy system decarbonisation challenge. The TIMES-Ireland Model (TIM) is an optimisation model of the Irish energy system, which calculates the cost-optimal fuel and technology mix to meet future energy service demands in the transport, buildings, industry and agriculture sectors, while respecting constraints in greenhouse-gas emissions, primary energy resources and feasible deployment rates. TIM is developed to take into account Ireland's unique energy system context, including a very high potential for offshore wind energy and the challenge of integrating this on a relatively isolated grid, a very ambitious decarbonisation target in the period to 2030, the policy need to inform five-year carbon budgets to meet policy targets, and the challenge of decarbonising heat in the context of low building stock thermal efficiency and high reliance on fossil fuels. To that end, model features of note include "future proofing" with flexible temporal and spatial definitions, with a possible hourly time resolution, unit commitment and capacity expansion features in power sector, residential and passenger transport underpinned by detailed bottom-up sectoral models, cross-model harmonisation and soft-linking with demand and macro models. The paper also outlines a priority list of future model developments to better meet the challenge of deeply decarbonising energy supply and demand, taking into account equity, cost-effectiveness and technical feasibility. To support transparency and openness in decision-making, TIM is available to download under a Creative Commons licence.

## 1 Introduction

Ireland faces very significant challenges in meeting greater energy needs in the future with a much lower carbon footprint. Ireland has a high per-capita carbon footprint relative to the European average and will fail its 2020 decarbonisation objective as set by the European Union (EU) (DCCAE, 2019). Under existing policy measures, overall GHGs are projected to be relatively





stable in the period to 2030 and to increase in the period to 2040 (EPA, 2020). In contrast, the EU's Effort Sharing Regulation sets forth a legislated target which increases the Irish decarbonisation objective, to reduce non-emissions traded sector (ETS) emissions by 30% relative to 2005 levels by 2030 (CCAC, 2020). In 2019 the Government presented a Climate Action Plan
which set forth sector-by-sector measures to meet this increased ambition from the EU, which includes increasing the renewable electricity share to 70% by 2030, for electric vehicles to reach full market share later in the decade, and very ambitious targets for retrofitting and electrifying home heating.

However, additional policy measures are needed to reduce emissions even faster. In 2020, a new government adopted an even more ambitious decarbonisation target, to reduce emissions by 7% annually in the period to 2030, more than halving
emissions, as well as planning for a legislated "net-zero" emissions target by 2050 (Department of the Taoiseach, 2020).

Ireland faces a number of challenges in meeting these objectives. Firstly, a very high share of GHG emissions (34% in 2018 (Duffy et al., 2019)[1]) in Ireland arise in the agricultural sector, which is a large and export-led part of the economy, dominated by beef and dairy production, with an emissions profile which is considered more difficult to abate than energy sectors. Slower mitigation in this sector will require energy to decarbonise faster. Secondly, transport and heating are heavily dependent on
fossil fuels (with shares of 94% and 96% of consumption respectively) (SEAI, 2019), while dispersed settlement patterns and an inefficient building stock make improve efficiencies challenging. Thirdly, while Ireland has already been successful in integrating 36.5% of renewable electricity into power generation, 86% of which is from wind energy, the relatively isolated nature of the electricity grid and lack of alternative low-carbon electricity sources will make it very challenging to integrate high shares of renewable electricity. The TIMES-Ireland Model (TIM) has been built to offer mitigation solutions taking these
challenges into account (Balyk et al., 2021).

Energy system models have long been used to inform decarbonisation policies both in Ireland and other countries. Integrated and dynamic energy systems models have a number of advantages over single-sector or static approaches. Current energy systems are the result of complex country-dependent, multi-sector developments. The complex dynamics (incorporating technologies, fuel prices, infrastructures and capacity constraints) of the entire energy system can be analysed through this
modelling approach to better inform policy choices. A key strength is to approach energy as a system rather than as a set of discrete non-interactive elements. This has the advantage of providing insights into the most important substitution options that are linked to the system as a whole, which cannot be understood when analysing a single technology, commodity or sector. A single focus on the electricity sector, for example, risks excluding possible unforeseen step changes in electricity demand, because of, perhaps, the electrification of transport or of heating.
TIM is the successor to the Irish TIMES model (Ó Gallachóir et al., 2020), which has a long (more than 10-year) history of providing analytical input to Irish energy policy development, including acting as the basis for Ireland's first Low-Carbon Roadmap in 2015 (Deane et al., 2013) and for developing energy pathways consistent with the Paris Agreement (Glynn et al., 2019), to which Ireland is a signatory. TIM is a new model and has been developed to better inform increased national climate

---

[1]These emissions are mainly non-$CO_2$ emissions, $CH_4$ and $N_2O$, arising from enteric fermentation, fertiliser application and soil management, and does not include emissions from land-use, while grasslands are a net carbon source





mitigation ambition, to take into account the changing energy technology landscape, and to take advantage of new advances in
energy systems optimisation modelling techniques, which are described in the remainder of this paper.

Internationally, TIMES models are used in a number of countries to understand and plan for long-term energy transitions,
including in Denmark (Balyk et al., 2019) and the United Kingdom (Fais et al., 2016; Daly et al., 2015).

Other energy system models in Ireland which are used to inform long-term pathways include the LEAP-Ireland model
(Mac Uidhir et al., 2020; Rogan et al., 2014), based on a simulation approach, which is being co-developed with TIM to
take advantages of data harmonisation and complementary of policy insights. Furthermore, the Economic and Social Research
Institute (ESRI) develops the I3E model, a top-down computable general equilibrium (CGE) model (see Section 2.6). The 2019
Climate Action Plan was informed by McKinsey's Marginal Abatement Cost Curve (MACC) (DCCAE, 2019).

TIM is a significant step forward in national energy systems modelling capacity. A number of features better enable this
model to capture long-term energy systems transitions, enabling it to better inform very ambitious decarbonisation targets.
These features are described in detail throughout the paper and are summarised in the Discussion (Section 4).

The rest of the paper is structured as follows: Section 2 gives a general description of the model, including a "plain En-
glish" description of the model (2.1), an outline of the TIMES methodology and model generator (2.2), an overview of the
system (2.3), the temporal and regional characteristics (2.4), underlying demand drivers (2.5), and the model development
approach (2.6). Section 3 describes the model sectors (Supply, Power, Transport, Residential, Services and Industry). Section
4 discusses model strengths, weaknesses and priority areas for future development. Finally, Appendix A includes additional
techno-economic assumptions for future technologies.

## 2 Model description

### 2.1 Plain English description

The TIMES-Ireland Model produces energy system pathways for energy supply and demand in Ireland consistent with either
a carbon budget or a decarbonisation target. It calculates the lowest-cost configuration of energy fuels and technologies which
meet future energy demands, while respecting technical, environmental, economic, social and policy constraints. Key inputs
and constraints include primary energy resource availability and costs, the technical and cost evolution of new mitigation
options and maximum feasible uptake rates of new technologies. Alternatively, TIM can be used to assess the implications of
certain policies, namely regulatory or technology target-setting (for example, biofuels blending obligation or sales/stock share
target for electric vehicles).

### 2.2 TIMES Model generator

TIMES (The Integrated MARKAL-EFOM System) is a bottom-up optimisation model generator for energy-environment sys-
tems analysis at various levels of spatial, temporal and sectoral resolutions (Loulou et al., 2016a, b). The TIMES code, written
in GAMS and available under an open source licence (IEA-ETSAP, 2020) is developed and maintained by the Energy Tech-





nology Systems Analysis Programme (ETSAP)[2], a Technology Collaboration Programme (TCP) of the International Energy Agency (IEA), established in 1976. TIMES models can have single or several regions, and typically are rich in technology detail, used for medium- to long- terms energy system analysis and planning at a regional, national or global scale.

TIMES is a linear optimisation, techno-economic, partial-equilibrium model generator which assumes perfectly competitive markets and perfect foresight. Model variants enable myopic foresight, general equilibrium, stochastic programming and a variety of multi-objective function options. The standard objective function maximises the net total surplus (the sum of producers' and consumers' surpluses) which, in a perfect market with perfect foresight, equates to maximising the net present value (NPV) of the whole energy system, maximising societal welfare. Profits, taxes and subsidies are internal transfers that do not change the NPV. It calculates the energy system specification which minimises discounted total energy system costs over the model time horizon, which is the sum of investments, fixed and variable costs, fuel import costs and export revenues for all the modelled processes, less potential salvage values of investments whole lifetime goes beyond the model time horizon.

The user inputs the following to the model generator:

– Reference Energy System (RES), is the process-flow architecture of economic sectors and energy flows (commodity) between processes (technology), which consume and produce energy, energy service demands and/or other commodities such as environmental emissions (including greenhouse-gases) and other materials. The base-year energy flows are calibrated to national energy balances.

– Energy service demands, are the physical services required by the economy and society for mobility, heat, communications, food etc., which drive energy demand.

– Energy supply curves, the quantities of primary energy resources (e.g. wind power) or imported commodities (e.g. oil, gas, bio-energy) available at specific costs points for differing quality and quantity of energy commodities.

– Techno-economic parameters of existing and potential future energy technologies: economic parameters including current and projected future investment and fixed/variable costs and efficiencies of technologies for energy supply (e.g., solar PV panels, transmission and distribution infrastructure, biorefineries, hydrogen production) and energy demand (e.g., electric vehicles, natural gas boilers, carbon capture and storage); technological parameters including transformation efficiency, availability factor, capacity factor and emissions factor.

– User constraints, which can be any combination of linear constraints (including fixed, maximum or minimum bounds on growth, investment or shares) on technologies or fuels. These are typically used to simulate real-world technology constraints or to simulate policy scenarios. A typical user constraint for decarbonisation analysis is limiting total annual or cumulative $CO_2$ emissions to model energy system pathways to meeting a national decarbonisation target.

TIMES outputs the optimal investment and operation level of all energy technologies which meet future energy service demands at least cost, while respecting user constraints. The model also produces corresponding energy flows, emissions and marginal prices of energy and emissions flows.

---

[2]https://iea-etsap.org/





## 2.3 Model architecture

Fig. 1 shows a simplified RES in TIM. It describes the structure and energy flows including two major parts: supply-side and demand-side. The former comprises energy resources, fuel production and conversion technologies (e.g. biorefineries, hydro-
gen production and different power plants), transmission and distribution infrastructure (e.g. gas pipelines and power grid). The latter covers end-use sectors (e.g. transport, residential) and the corresponding energy service demands (i.e. passenger, freight, hot water). Energy resources incorporate both domestic fossil-based fuels and renewables potentials. These fuels are processed and then distributed across country. End-use technologies consume energy commodities to meet energy service demands. GHG emissions from fossil fuels combustion and process-related emissions in industry are tracked at the fuel supply
module, electricity generation technologies and sectoral-consumption levels.

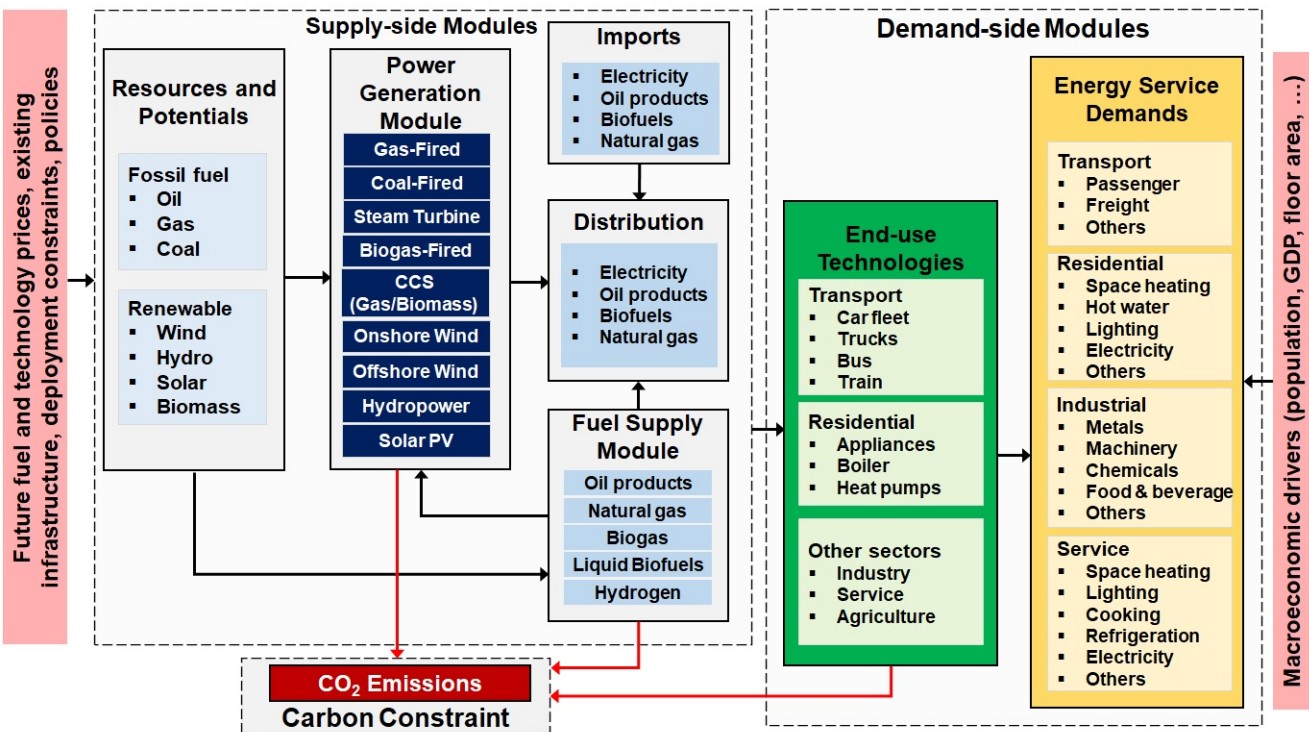

**Figure 1.** Simplified representation of reference energy system in TIM

The model's base year is 2018 and all energy flows, emissions and energy technology stocks are calibrated to the 2018 energy balance (SEAI, 2019).

The discount rate, the degree to which future values are discounted to the present, is a key parameter in the TIMES objective function. A social discount rate reflects how society views present costs and benefits against future ones and is lower than
a financial discount rate, which is how firms make investment decisions. In appraising potential projects or investments, the Government applies a social discount rate. Broadly speaking, in an ESOM scenario with a carbon budget constraint, a higher





discount rate would promote later decarbonisation and less capital-intensive technology choices. In this model, a discount rate of 4% is applied, which is based on a Social Rate of Time Preference methodology as set forth in the Public Spending Code (O'Callaghan and Prior, 2018). This rate is consistent with García-Gusano et al. (2016) who recommend using a maximum

value of 4-5% for the social discount rate in ESOMs.

Technology-specific discount rates (also known as hurdle rates) are typically used in ESOMs to capture investment decision-making from the individual user or industry perspective, to capture market imperfections, limited finance and behavioural heuristics which limit the uptake of novel or capital-intensive investments, even when they are cost-optimal. These parameters are not used in the core version of TIM given its use for modelling long-term energy system pathways from a societal perspec-

tive. However, future variants of the model can be developed to simulate real-world impacts of policies and behaviour which can include hurdle rates.

## 2.4 Time & Geography

TIM has been developed with a deep knowledge of the geography of the Irish energy system. A spacial spatio-temporal approach was taken in the RES base year specification and scenario file data structures to allow flexible regional definitions

and temporal resolution in TIM. The model can run in multiple modes with multiple configurations of regional and temporal resolution ranging from a single region national model at one annual time slice, all the way to 26 counties at hourly resolution where supply-demand data is available at that spatio-temporal granularity (Electricity, Gas & Transport). Furthermore within the power sector each existing individual power plant turbine is represented individually with a possibility to use unit commitment. High techno-temporal granularity is needed to appropriately model energy futures with high variable renewable energy

systems integration, especially in scenarios with high levels of electrification of end use demands. High spatial granularity is required to give greater policy clarity on optimal investment needs based on regions and counties specific characteristics to enable counteracting socio-economic challenges such as energy poverty and infrastructure development within an optimisation framework. We have decided to focus on this data-driven spatio-temporal model setup to future-proof TIM and enable future Irish energy policy research needs as data-availability improves during TIM's lifespan.

## 2.5 Demands: Drivers & projections

Energy service demands in end-use sectors are driven by growth in the population and in the economy. The model is set up to allow for alternative scenarios for these drivers resulting in different energy service demand projections in the end-use sectors. This section describes baseline driver projections with a detailed description of energy service demand projections included in the respective sector sections.



### 2.5.1 Population

Historical population estimates and future projections are obtained from CSO (2020d). We use the M2F1 scenario since it represents a medium growth in population and is in line with population projections used in other national sources (Yakut and de Bruin, 2020).

**Table 1.** Population

| Year | Population (millions) |
|------|-----------------------|
| 2018 | 4.85 |
| 2020 | 4.98 |
| 2030 | 5.40 |
| 2040 | 5.82 |
| 2050 | 6.19 |

### 2.5.2 Economic growth

- Historical Gross Value Added (GVA) for the required NACE categories in the Services and Industry sectors is obtained from EUROSTAT database. Projections for GVA are outputs of the Ireland Environment, Energy and Economy (I3E) model (Yakut and de Bruin, 2020).

- Gross Domestic Product (GDP), both historical and future projections, is obtained from OECD (2018).

- Income, historical values of total incomes, are taken from CSO (2021). Assumption about income growth in the future are from the National Transport Model (AECOM, 2019).

- Modified Gross National Income (GNI*) is derived from CSO's labour force scenario combined with a forecast for output per person (CSO, 2018).

### 2.5.3 Energy service demands

Table 2 lists the energy service demands in TIM along with their corresponding drivers and values for 2018 and 2050. Specifics of the methodologies for projecting each energy service demand are detailed in later sections, for transport (3.3), residential (3.4), industry (3.5) and services (3.6).

### 2.6 Development approach

TIM has been developed with the goal of achieving "best practice" standards in software development and open modelling convention. A git-centred model development process has been an integral part of the model development approach to enable version control and model management. Along with improvements in management, quality assurance and transparency this brings, it also allows developers and researchers from different projects to branch research versions of the model, to explore



**Table 2.** Energy service demands in TIM

| Energy Service Demand | Driver / Projection source | Value 2018 | Value 2050 | Unit |
|---|---|---|---|---|
| Non-Energy Mining | GVA per capita, Population | 2.07 | 0.13 | PJ |
| Food and beverages | GVA per capita, Population | 22.25 | 34.00 | PJ |
| Textiles and textile products | GVA per capita, Population | 1.20 | 4.97 | PJ |
| Wood and wood products | GVA per capita, Population | 6.69 | 7.65 | PJ |
| Pulp, paper, publishing and printing | GVA per capita, Population | 0.67 | 2.31 | PJ |
| Chemicals and man-made fibres | GVA per capita, Population | 10.60 | 13.11 | PJ |
| Rubber and plastic products | GVA per capita, Population | 1.14 | 0.89 | PJ |
| Other non-metallic mineral products | Modified investment, GNI* | 17.77 | 24.82 | PJ |
| Basic metals and fabricated metal products | GVA per capita, Population | 19.54 | 21.73 | PJ |
| Machinery and equipment n.e.c. | GVA per capita, Population | 1.29 | 1.69 | PJ |
| Electrical and optical equipment demand | GVA per capita, Population | 4.27 | 16.37 | PJ |
| Transport equipment manufacture | GVA per capita, Population | 0.17 | 0.04 | PJ |
| Other manufacturing | GVA per capita, Population | 4.25 | 6.12 | PJ |
| Construction | GVA per capita, Population | 4.02 | 5.90 | PJ |
| Transport Demand: Short-range passenger travels | Income, population | 14.56 | 21.07 | Bpkm |
| Transport Demand: Medium-range passenger travels | Income, population | 31.28 | 45.29 | Bpkm |
| Transport Demand: Long-range passenger travels | Income, population | 27.13 | 38.97 | Bpkm |
| Transport Demand: Goods vehicle for freight | Growth rate (AECOM, 2019) | 11.54 | 25.14 | Btkm |
| Transport Demand: Turism fuel | | 7.72 | 0.00 | PJ |
| Transport Demand: Navigation fuel | GDP | 3.51 | 10.04 | PJ |
| Transport Demand: Unspecified fuel | | 21.78 | 0.00 | PJ |
| Transport Demand: Aviation domestic | | 0.23 | 0.23 | PJ |
| Transport Demand: Aviation international | International Aviation Passengers | 45.94 | 62.63 | PJ |
| Residential Apartment Demand | Population | 206.80 | 628.93 | 000' |
| Residential Attached Demand | Population | 766.35 | 1056.50 | 000' |
| Residential Detached Demand | Population | 724.43 | 889.76 | 000' |
| Services - Commercial Services | GNI* | 28.90 | 47.39 | $Mm^2$ |
| Services - Public Services | GNI* | 58.15 | 95.35 | $Mm^2$ |
| Services - Commercial Services - Data centers | EirGrid (2017) | 5.63 | 40.30 | PJ |
| Services - Public Services - Public lighting | Government of Ireland (2018) | 0.48 | 0.58 | Mlamps |





innovations and new developments, while keeping a secure and stable main version of the model for policy application. At the same time, individual projects and researchers can input their improvements and developments to the core model, to enable continuous improvements.

TIM is freely available, which is a prerequisite for transparency, repeatable research, model maintenance and enhancement and verification of results (Pfenninger et al., 2018).

Web-based dashboards[3] have been extensively used in the model development process, both for internal model diagnostics and for external engagement and review. The first TIM scenario results archive has also been published on Zenodo (Daly et al., 2021).

TIM has been co-developed with the LEAP-Ireland model (Mac Uidhir et al., 2020), which is a bottom-up simulation model of the Irish energy system which simulates the impact of different policy measures and targets on overall GHG emissions in Ireland, with a particularly granular representation of transport and residential heat demand. Underlying data for the relevant sectors are shared between TIM and LEAP using a Data Repository and shared coding convention dictionary to improve the consistency between the models, give more robust analytical insights for policy and to share and exchange expertise between the

modelling teams. This also facilitates multi-model approaches to energy systems modelling, which can make use of harmonised hybrid frameworks coupling simulation and optimisation simultaneously (Rogan et al., 2014).

TIM has also been developed to enable soft-linking with the Ireland Environment, Energy and Economy (I3E) macroeconomic model developed at the Economic and Social Research Institute (ESRI) (Yakut and de Bruin, 2020). I3E is a single-country, multi-sector (NACE) inter-temporal computable general equilibrium (CGE) model focusing on environmental and

energy accounts in Ireland. While COre Structural MOdel for Ireland (COSMO) focuses on the influence of monetary and fiscal policy on economic activity in Ireland, I3E supplements the macroeconomic outlook from COSMO with environmental and energy disaggregation. I3E retrieves economic growth rates and population estimates from COSMO. TIM derives macroeconomic drivers coupled to the output variables of I3E, enabling scenario variants based on alternative monetary, fiscal and macroeconomic futures, as well as rapid energy system outlook updates aligned with the update cycle of the macroeconomic

outlooks from the ESRI.

## 3   Sectors

### 3.1   Supply

#### 3.1.1   Overview

The supply sector (SUP) in TIM represents the primary and secondary energy commodities and the processes by which those

same commodities are imported, exported, domestically produced through mining or capture of renewable energy potentials and transformed or refined for end-use consumption within the energy system both in the base year (2018) and into the future. The supply sector declares the future available routes for commodity trade for import and export of energy commodities in

---

[3]https://tim-review1.netlify.app/results/





terms of quantity of energy, and in terms of import capacity through ports, pipelines and inter-connectors at any given time in the model horizon.

### 3.1.2 Energy Balance and Commodity Declarations

Building the supply sector begins with declaring the energy commodities as per the SEAI (2019) energy balance, as reported to the International Energy Agency. Attention is taken to ensure best practice coding conventions are followed for each commodity - coal, oil, gas, first- and second-generation bioenergy (biogas, bioliquids and solid biomass), liquid and gaseous hydrogen, wind, solar, geothermal, wave, tidal, municipal wastes, agriculture wastes, industrial food waste. Active transport time for walking and cycling is also declared. Setting out predefined and intuitive commodity naming convention and a dictionary that is shared across TIM and LEAP-Ireland has multiple benefits for multi-model coupling, diagnostics and results reporting that is discussed. All base year commodities are declared in the Supply sector to maintain clear, tidy and transparent data structures within TIM.

### 3.1.3 Emissions Tracking

The environmental emissions from each primary energy commodity are tracked on the basis of energy and processes-based emissions via combustion and activity-based emissions intensity factors, calibrated on an sector-by-sector basis. Within the Supply sector, methane, nitrogen dioxide, sulfur dioxide, carbon dioxide, nitrogen oxide, and particulate matter can be tracked in the sector's emissions accounting balance.

### 3.1.4 Import/Export

Primary and secondary energy commodities, both fossil energy and bioenergy, are imported from international markets at international prices. There are no constraints on the import quantity of oil and coal as it is assumed that international markets can supply domestic demand and that there is sufficient on island storage aligned with IEA-OECD energy security protocols. Imported gas via pipeline and LPG are modelled on an annual basis. Bidirectional electricity inter-connectors to the UK are also represented.

### 3.1.5 Fuel prices

Fuel prices for each imported energy commodity are sourced from IEA (2019, 2020) World Energy Outlook. Secondary commodity import prices are index linked to the primary commodities by a price ratio on the basis of the current ratio between primary and secondary commodity prices. For example, imported gasoline is assumed to be 1.65 times the price of crude oil per unit energy.





### 3.1.6 Domestic Energy Resources

Domestic fossil fuel reserves for production of natural gas from both the Corrib and Kinsale gas fields, and the production of peat, are calibrated with two-step supply curves. These supply curves are constrained in terms of cumulative energy reserves, annual production costs, and annual production output in energy terms to account for typical production profiles from each field.

Renewable energy potentials (hydro, wind, solar, waste, ocean, geothermal, and ambient heat) are declared in the supply sector but are calibrated and constrained in their relevant sectors, such as power generation and transport.

### 3.1.7 Bioenergy potentials

Bioenergy potentials are calibrated to SEAI (2015) both in terms of sustainable import volume availability and domestic production potentials. Domestic bioenergy potentials such as sawmill residues, post-consumer recycled wood, municipal waste, tallow, recovered vegetable oil, straw, animal waste and industrial food wastes are modelled with three-step supply curves in terms of price and quantities available. Crop-based bioenergy feedstocks are modelled within the agriculture sector. Domestic bioenergy supply potentials and costs are sensitive to scenarios narrative and as such are modelled within scenario files to account for uncertainty.

Some agriculture sector commodities (i.e. bioenergy, land availability and herd types) are declared in the supply sector.

### 3.1.8 Refineries

Ireland has only one oil refinery: Whitegate, in County Cork. It is calibrated to import crude oil at international prices and converts crude oil to refined products limited by upper bounds of output shares such that the output has flexible upper shares of 22.9% gasoline, 8.1% kerosene, 40.9% diesel and 34.4% heavy fuel oil. The refinery is constrained to stay at current capacity, has a lifespan of 50 years, and the production costs are differentiated by production (flow) costs for each output fuel.

### 3.1.9 Electricity Inter-connectors

Existing electricity inter-connectors to the UK are calibrated such that exports are priced 30% lower than imports. The 2018 export and import quantities of electricity are calibrated as per historical data. The future activity is constrained using upper bounds, given that TIM does not model the UK electricity market, other than through predefined price signal. This constrain can be relaxed, but is used to explore domestic needs for system flexibility, security, system services and storage.

### 3.1.10 Biorefineries

Future bio-refinery technologies are defined in the supply sector future technology (SubRES) database. The following technologies are defined: 1) ethanol production from wheat, woody biomass and grass; 2) bio-diesel production from OSR, woody biomass, industrial food waste, RVO and tallow; 3) wood pellets production from biomass; and 4) biogas production from





grass, woody biomass, municipal waste, industrial food waste and animal wastes. All bio-refining technologies are defined in
terms of start year, efficiency, investment costs, availability factor, and the operation and maintenance costs.

### 3.1.11   Hydrogen

Hydrogen production is modelled in the future, disaggregating centralised and decentralised electrolysis options. Delivery options are disaggregated and costed at high pressure for both transmission and distribution pipelines as well as a road tanker option for distribution. Hydrogen storage is also modelled within TIM at the DAYNITE timeslice level allowing hourly pro-
duction and consumption of hydrogen to be represented. This is particularly useful for modelling electricity grid balancing with unit commitment, dispatch and capacity expansion while capturing variable renewable energy system dynamics.

### 3.2   Electricity

### 3.2.1   Overview

Ireland has a high share of variable renewable electricity for a relatively isolated grid, with 32.5% of electricity generation in
2019 coming from onshore wind energy. Achieving the 2020 RES-E target has encouraged strong growth in onshore wind, while increasing the non-synchronous penetration of renewables to 70% by 2030 including offshore wind development is a key policy objective over the next decade as Ireland moves towards a net zero carbon electricity system.

### 3.2.2   Existing Dispatchable Grid

The existing 61 generation unit fleet is calibrated to the base year of 2018 from the Commission for Energy Regulation (CRU)
I-SEM validated PLEXOS model (Geffert et al., 2018). Each existing generation unit can be explicitly modelled with unit commitment within TIM. This model configuration includes the generation unit capacity, the fuel type of each generation unit, the start up costs (from cold, warm and hot), the efficiency curve of the generation unit to include startup and shutdown phases, startup and shutdown times (cold, warm and hot), the ramp rates, minimum load, efficiency at minimum load, minimum uptime and minimum down-time, the unit lifespan, the unit annual availability, and the start year of the unit (Geffert et al., 2018).
The near-term generation unit pipeline is calibrated to EirGrid and SONI (2019, 2020) to account for early closures of units before their economic lifespan, which largely takes into account coal and peat based plants during the next decade. We have not forced on new capacity of future planned power plants from EirGrid and SONI (2020) in TIM, which includes recent battery storage installations. These recent and planned units can be forced on via a scenario file.

The existing 302 Onshore and 1 Offshore Wind farms within the Irish Wind Energy Association database currently operate
as a single aggregated generation pool governed by historical hourly availability factors for wind generation from 2018.



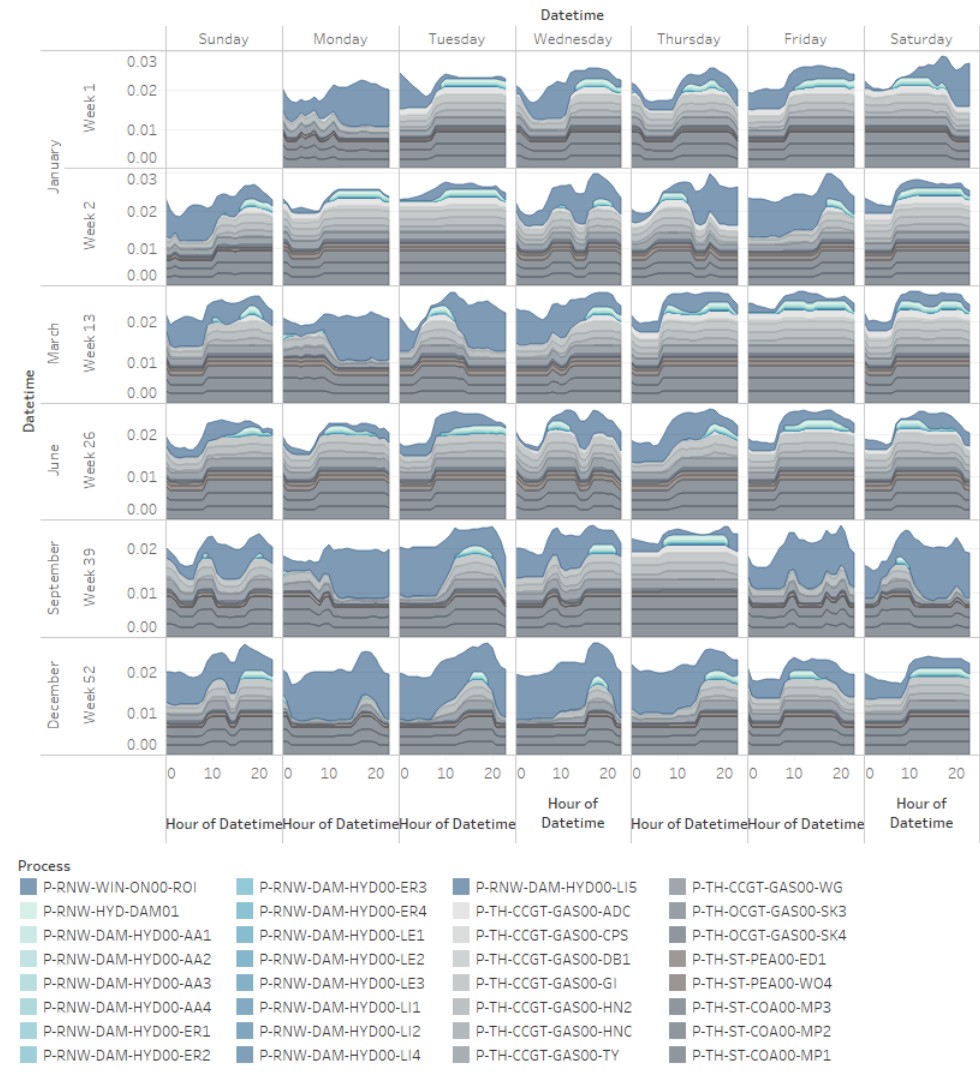

**Figure 2.** TIM hourly unit commitment modelling capabilities example for 2018





### 3.2.3 Future Electricity system

New generation capacity investment costs, fixed and variable operation and maintenance costs and technical efficiencies for 2010-2050 are derived from Carlsson et al. (2014). The future unit commitment dispatch operational constraints and cycling costs are generalised for each technology type based on fuel and vintage (Kumar et al., 2012).

Renewable energy potentials are based on a number of sources. Total energy resource availability of solar and wind are from Pfenninger and Staffell (2016) and Staffell and Pfenninger (2016) respectively with availability factors from Ruiz et al. (2019). Ocean and tidal energy potentials are from O'Rourke et al. (2010), and wave energy power matrix is derived from Nambiar et al. (2016). Hourly availability factors for ocean energy technologies are derived from the marine institute and OPW's data buoy services.

Future onshore wind energy potential has been assessed at high spatial resolution using GIS techniques. Wind farm expansion is constrained by both technology costs and a supply curve for grid connections based on a GIS analysis of existing houses, special areas of heritage and conservation, existing grid lines by voltage specification, wind energy potential, existing substation locations, and the costs per kilometre of standard ESB grid line specifications to connect wind resource potential to the existing grid. The wind energy potential is defined as a cost curve aggregated from 1938 suitable land parcels from a GIS analysis (Fig.

3). Due to the fact that increasing both technical details and temporal resolution causes exponential increase in model size, the spatial resolution is simplified and is represented in a 4 step cost curve (Table 3). This can be dissaggregated on a spatial resolution down to each individual parcel of land, or on a county by county basis. Future wind energy can generate at the same hourly capacity factor that occurred in 2018, however with capacity expansion.

Losses from electricity transmission and distribution are assumed to be 7%, with no grid expansion costs currently repre-

sented in the model. The maximum share of variable renewable energy including wind, solar and wave energy is constrained at 75% by 2030 and 100% is allowed by 2050.

**Table 3.** Onshore wind connection cost

| Capacity (GW) | Connection Cost (EUR/kW) |
|---|---|
| 0-8.1 | 25 |
| 8.1-16.8 | 39 |
| 16.8-22.6 | 69 |
| 22.6-26.6 | 153 |
| 26.6-30.6 | 358 |

### 3.2.4 Carbon Capture and Storage, Carbon Dioxide Removal, and Negative Emissions technologies

It is well documented in the literature that residual emissions are likely to remain in the future (net) zero carbon energy system from "hard-to-decarbonise" sectors (Rogelj et al., 2018). Carbon Capture and Storage (CCS), Carbon Dioxide Removal

technologies (CDR), and Negative Emission Technologies (NETS) are currently seen within the literature as critical future technologies to capture the last marginal residual emissions to bring the energy system to net-zero $CO_2$ or even net-negative



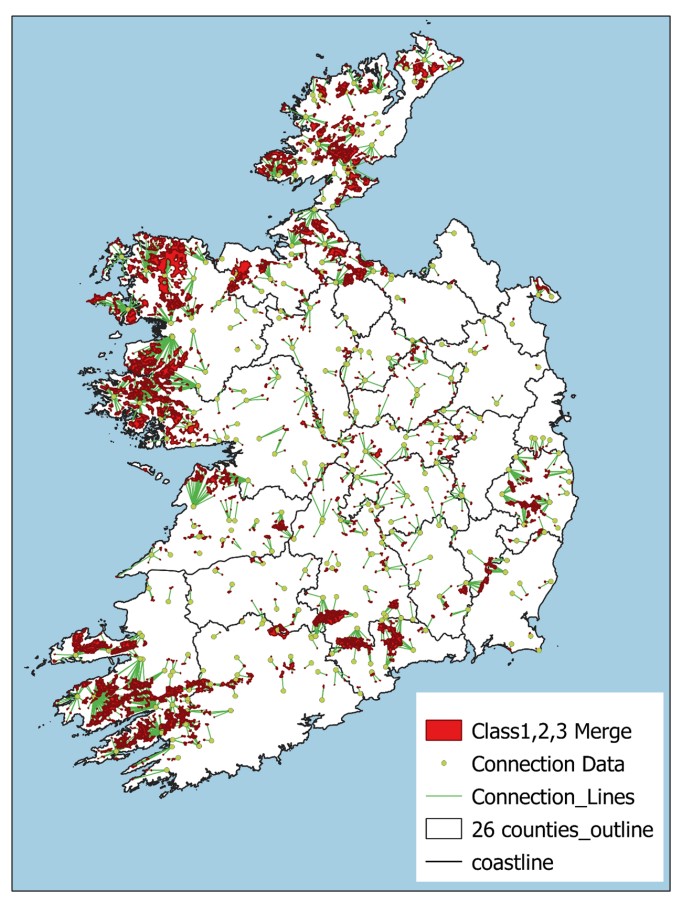

**Figure 3.** Wind energy potential locations

CO2 by mid-century. With this in mind, TIM has carbon capture and storage (CCS) technology options available from 2030, including retrofit options on existing coal, peat and gas power plants. Bioenergy carbon capture and storage (BECCS) is available in TIM from 2030 and provides net negative CO2 capture and allows negative emissions electricity generation. Direct air carbon capture and storage (DACCS) is also defined as a backstop technology i.e., it has a static unit cost of 2000 EUR/tCO2 and an unlimited capacity. This caps the marginal abatement cost of the model at a price that does not exclude any of the plausible mitigation measures.

### 3.2.5 User constraints

User constraints applied to the Power sector in TIM largely pertain to the calibration of the near-term shutdown of coal and peat power plants before their economic life-time ends as a result of policy decisions.





### 3.3 Transport

#### 3.3.1 Overview

The transport sector comprehensively describes the end-use transport technologies, and freight and mobility demands on a regional basis. This sector is divided into 26 counties across Ireland. To represent region-specific transport characteristics,
some main parameters (vehicle fleet, transport infrastructure, fuel consumption, mileage, occupancy rate, load factor) are differentiated on a county level. Transport demand is split to three main categories: passenger, freight, and others. The passenger and freight demands are expressed as activity demands, and others are defined as a final energy demand (PJ). These final energy demands further split into aviation (international and domestic), navigation, fuel tourism and unspecified, aligned with the energy balance (SEAI, 2019). "Fuel tourism" refers to cross-border consumers and a portion of demand is used by unspecified
modes.

The passenger transport demands are expressed in billion-passenger-kilometres (Bpkm). As shown in Table 4, the total passenger demand is divided into three classes of distance range including short-range (less than 5km), medium-range (5-30 km), and long-range (more than 30km) (NTA, 2018). Four transport modes satisfy travel demands including 1. public services (bus, train, taxi), 2. private cars, 3. powered two-wheelers (PTW), and 4. active modes (walk and bike). Non-motorised transport
is only used for short-range trips, PTW are used for short- and medium-range travels, urban bus and school bus are used for short- and medium range travels, Intercity bus and heavy train are used for long-range trips, and light rail can only be used for the short- and medium-range trips in Dublin county. As shown in Fig. 4, demand for each mode can be met with a different set of technologies based on cost-optimisation and user constraints. The base year is calibrated based on the actual number of vehicles and the corresponding vehicle activities. Table 6 shows vehicle characteristics on the national basis.

**Table 4.** Total passenger demand and share of transport modes for each class of distance range (CSO, 2017, 2020c, e, f, g, h)

| Modes | Vehicles | Short-range (below 5km) | Medium-range (5-30km) | Long-range (over 30km) |
|---|---|---|---|---|
| Public | Bus | 8.3% | 13.5% | 16.1% |
| | Light train | 0.8% | 0.7% | NA |
| | Heavy train | NA | NA | 8.4% |
| | Taxi | 1.7% | 2.2% | 1.3% |
| Private | Car | 51.5% | 83.3% | 74.2% |
| Powered two-wheelers | 0.1% | 0.3% | NA | |
| Active | Bike | 5.4% | NA | NA |
| | Walk | 32.2% | NA | NA |
| Total passenger demand in 2018 (Bpkm) | | 14.6 | 31.3 | 27.1 |

The inland freight demand is expressed in billion tonne-kilometres (Btkm). It comprises two main modes: goods trucks and train. The definition of light and heavy goods vehicles varies in different studies. In this model, they are disaggregated by three





unladen weight bands: light-duty trucks (below 5 tonnes), medium-duty trucks (5-10 tonnes) and heavy-duty trucks (over 10 tonnes) (CSO, 2020g, h, i). Table 5 shows freight demand in the base year in million tonne-kilometres (Mtkm).

**Table 5.** Freight demand in 2018 (CSO, 2020g, h, i)

| Classification | Unladen weight | Demand (Mtkm) | Share |
|---|---|---|---|
| Light-duty trucks | 0-5 tonne | 292 | 2.5% |
| Medium-duty trucks | 5-10 tonne | 1140 | 9.8% |
| Heavy-duty trucks | over 10 tonne | 10106 | 86.9% |
| Train | - | 89 | 0.8% |
| Total Freight demand | | 11627 | 100% |



**Figure 4.** Transport structure in TIM





**Table 6.** Existing vehicles and the corresponding characteristics in the base year (CSO, 2020j, k, l; Irish Rail, 2018; TII, 2016)

| Vehicles | Powertrain | Stock (000-Units) | Utilisation factor (1000km/yr) | Occupancy rate (Pass./Vehicle) | Fuel consumption (MJ/v.km) |
|---|---|---|---|---|---|
| Motorcycle | Gasoline ICE | 39.85 | 2.73 | 1.10 | 1.70 |
| Cars | Gasoline ICE | 946.86 | 12.82 | 1.49 | 2.47 |
| | Diesel ICE | 1129.40 | 20.62 | 1.49 | 2.30 |
| | Dual-fuel ICE | 0.07 | 13.44 | 1.49 | 2.89 |
| | ICE-E85 | 8.53 | 13.44 | 1.49 | 2.41 |
| | Gasoline HEV | 29.80 | 12.82 | 1.49 | 2.05 |
| | Diesel HEV | 0.77 | 20.62 | 1.49 | 2.03 |
| | Gasoline PHEV | 2.76 | 12.82 | 1.49 | 1.56 |
| | Diesel PHEV | 0.03 | 20.62 | 1.49 | 1.58 |
| | BEV | 4.53 | 13.44 | 1.49 | 0.85 |
| Taxi | Gasoline ICE | 2.50 | 35.61 | 1.49 | 2.63 |
| | Diesel ICE | 17.46 | 39.93 | 1.49 | 2.39 |
| | Gasoline HEV | 1.35 | 41.21 | 1.49 | 2.03 |
| Bus | Diesel-ICE | 10.70 | 36.10 | 27.25 | 10.16 |
| Train | Light train (Electric) | 0.07 | 55.69 | 78.0 | 24.81 |
| | Heavy train (Electric) | 0.05 | 158.48 | 78.0 | 24.81 |
| | Heavy train (Diesel) | 0.014 | 73.88 | 120.0 | 76.92 |

### 3.3.2 Future transport demand projections

### 3.3.3 Passenger-kilometres: Private Cars

Future passenger car transport demand is projected based on future population growth and a growing rate of car ownership, which is in turn determined by income growth. Car ownership usually follows an S-shaped function which has three periods: slow growth during low income levels, rapid increase as income levels rise quickly and finally a saturation period. Gompertz statistical model has been found to best fit the historical relationship between car ownership and income levels, although other

functions have also been used in previous studies (Lian et al., 2018). The basic Gompertz function is shown in Equation 1.

$$y = \alpha * e^{-\beta * e^{-\gamma x}} \tag{1}$$

where $y$ is the car per adult, $\alpha$ is saturation level of car ownership, $x$ is an economic indicator (income per adult in this case)

and $\beta, \gamma$ are parameters that are estimated using historical data obtained from CSO.

Projection of future car ownership levels is based on change in income levels. The saturation level of car ownership is assumed at 875 per 1000 adults (AECOM, 2019). Car ownership (cars per adult) is projected to rise from 0.56 in 2018 to 0.69 in 2050, an increase of 23%. Passenger-kilometres are then derived using car ownership as a proxy and assuming an occupancy





level of 1.492 and kilometres per car to remain constant at about 17300 per year. Total passenger-kilometres from private cars
in 2050 is projected to increase by 42% from the 2018 level with a Compound Annual Growth Rate (CAGR) of 1.1%. The
growth rate of passenger-kilometres from private cars was 1.35% between 2008 and 2018.

### 3.3.4    Passenger-kilometres: Other modes of transport

Other modes of transport represent a much smaller share of mobility demand compared to private cars. Passenger-kilometres
of large public service vehicles (PSVs) are projected using population as a driver in a log function and assuming average
occupancy of 27.5. Large PSV passenger-kilometres are expected to increase by 24.2% in 2050 as compared to that in 2018
with a CAGR of 0.7%. Passenger-kilometres of other modes (luas, train, small PSVs, and motorcycles) and active modes
(walking and cycling) are projected using population as a driver. The passenger-kilometres are expected to increase by 60%
with a CAGR of 1.5%.

### 3.3.5    International Aviation fuel demand

International aviation fuel demand is projected using number of passengers as a driver. The number of aviation passengers is
projected using damped Holt Winters function based on historical time-series data obtained from CSO (Dantas et al., 2017;
Grubb and Mason, 2001). The number of passengers in 2050 is expected to increase by 45.5% compared to 2018. The historical
fuel demand for aviation and number of aviation passengers are then used as input for a linear regression model to project the
future demand for aviation fuel. The fuel demand in 2050 increases by 37% relative to 2018 with a CAGR of 1%.

### 3.3.6    Other transport fuel demand

Demand for freight is projected using growth rates from AECOM (2019). The growth in tonne-kilometres of freight is expected
to increase by 1.18 times in 2050 from 2018 level with a CAGR of 2.5%. Navigation fuel demand is projected using GDP as the
explanatory variable. Fuel demand for navigation in 2050 is expected to increase 2.85 times compared to 2018 with a CAGR
of 3.3%. Fuel tourism is assumed to remain constant at 11 PJ.

### 3.3.7    Future technology options

Common vehicle technologies and future options that are likely to become available for future investment shape technology
database for the transportation sector in TIM. They are categorised in five major groups (Aryanpur and Shafiei, 2015):

1. Internal Combustion Engines (ICEs), consists of spark ignition engines fuelled by gasoline, bioethanol, CNG, BioCNG,
   hydrogen and dual-fuel engines (running either on gasoline or CNG/BioCNG, each one taking 50% of the distance
travelled), and compression ignition engines powered by diesel and biodiesel.

2. Hybrid Electric Vehicles (HEVs) are equipped with an ICE, which provides the main power, and a small electric motor
   to support the ICE and to recuperate the braking energy.





3. Plug-in Hybrid Electric Vehicles (PHEVs) have a similar powertrain to HEVs. Their batteries can be charged from the grid for driving tens of km solely on electrical power. We assume the maximum distance driven on electric mode is 50% in the base year and it can increase to 80% over time.

4. Battery Electric Vehicles (BEVs) solely rely on batteries, which provide the total motive power of the vehicle. The batteries are charged from electricity grid.

5. Fuel Cell Vehicles (FCVs) are electrochemical devices that produce electricity through a reaction between hydrogen and oxygen. The electricity drives a vehicle's electric motor. Conventional fuel tank is replaced with a pressurised hydrogen storage tank in FCVs.

Table A7 shows techno-economic characteristics of future passenger transport vehicles (Mulholland et al., 2017; Helgeson and Peter, 2020). Maintenance costs are assumed to remain constant. However, in some scenarios vehicle purchase price parity between BEVs and ICEs is expected in the period 2025-2030.

All these technologies compete to meet the mobility demand over the planning horizon. The model structure allows competition among stock replacement and fuel substitution within a mode. Modal shift may be simulated within each travel distance band.

Different fuels are supplied to the transport sector via four separated modules including supply, power, bio-refineries, and hydrogen modules. In other words, these connections integrate the transport sector with the entire energy system.

TIMES models usually use a simplified constant lifetime for different vehicles and thus, the vehicles are retired at the end of that lifetime. However, a detailed analysis of technology retirement profiles in Ireland shows that this simplified representation is far from reality (Mulholland et al., 2018). An actual profile shows a low decay in the beginning years and a long tail in the distribution over long-time. To improve the retirement profile both for existing and new vehicles, TIM is equipped with realistic representation of the survival profile of car technologies. The survival rates are from Irish CarSTOCK model (Daly and Ó Gallachóir, 2011; Mulholland et al., 2018).

### 3.3.8 User constraints

A set of constraints limits fuel and modal shares in transport sector as follows:

– Maximum biodiesel share in passenger and freight transport demand is 4.1% in the base year.

– Maximum bioethanol share in passenger and freight transport demand is 3.2% in the base year.

– Modal share of passenger and freight transport as stated in Table 4 and Table 5, respectively.

– Maximum growth rate in new vehicle sales for advanced powertrains (HEVs, PHEVs, BEVs and FCVs) is 16% per year, which is enforced once the sales of a vehicle type reaches 15% of the total vehicle sales in the base year.

– The blend limit for biofuels is 10-12% for the regular ICEs without any modifications (i.e. 10% bioethanol and 12% biodiesel with 90% and 88% gasoline and diesel, respectively).





– Total biofuel supply for the transport sector is allowed to double each decade, reflecting the rate of growth between 2010
and 2020 (NORA, 2021).





### 3.4 Residential sector

#### 3.4.1 Energy service demands and projections

The residential stock projections up to 2040 are taken from Bergin and García-Rodríguez (2020) housing demand estimates. The stock is expected to increase by 40% from 2018 level with a CAGR of 2%. This results in an average of 27,600 new

houses per annum between between 2021-2040. Beyond 2040, population is used as a driver to project housing stock. The total housing stock obtained in 2050 is 2.57 million dwellings which implies 8% increase from 2040.

**Table 7.** Number of dwellings by type ('000)

| Year | Apartment | Attached | Detached |
|------|-----------|----------|----------|
| 2018 | 207 | 766 | 724 |
| 2030 | 355 | 918 | 833 |
| 2040 | 493 | 1003 | 878 |
| 2050 | 629 | 1057 | 890 |

Fig. 5 shows the RES diagram for the residential sector. Energy service demands are disaggregated between archetype and non-archetype demands. "Archetype demands" are energy service demands which depend on the house type, namely the dwelling type and Building Energy Rating (BER) rating.

The base-year residential energy demand by fuel is calibrated to the SEAI (2019) energy balance.

Archetype energy service demands are particularly dependent on the type of building. The four energy service demands which are modelled based on archetype are: space heating, water heating, pump & fans and lighting. The residential building stock by type is explicitly modelled in TIM in three archetypes: detached, attached and apartments.

The Archetype energy service demand data are sourced from SEAI (2020) BER database, which contains the raw data of

906,048 BER surveys. The BER database was filtered before use to remove outliers and any nonsensical values. The filters were based upon Dineen et al. (2015), Uidhir et al. (2020a) and group discussions within the National Modelling Network Ireland reducing the total records to 815,246.

The filtered BER database is projected onto the total dwelling stock, using data from CSO (2020a, b) to calculate the total number BER ratings per archetype in the dwelling stock, as shown in Table 8.

BER assumes all buildings are heated to between 18°C for non-living areas (e.g. bedroom, bathrooms) to 21°C for living areas (e.g. sitting room, kitchen). This assumption is based on ISO 13790 calculations. To reflect actual energy use based upon internal temperatures in the Irish residential sector, the Archetype Dwelling Energy Model (ArDEM) (Dineen et al., 2015) was used to provide simulated annual energy consumption. ArDEM modifies the expected space heating energy consumption of each archetype and BER rating to the actual space heating energy consumption by adjusting internal temperatures in the

building stock, a similar approach was used by Uidhir et al. (2020b).

The alternative internal temperatures assumptions in TIM are outlined in Table 9.





**Table 8.** Residential Dwelling Stock in 2018

| BER Rating | Apartment | Attached | Detached | Total |
|---|---|---|---|---|
| A | 9,419 | 15,472 | 20,379 | 45,270 |
| B1 | 7,459 | 9,538 | 9,434 | 26,431 |
| B2 | 14,042 | 17,545 | 20,091 | 51,678 |
| B3 | 19,924 | 49,769 | 53,466 | 123,160 |
| C | 58,905 | 282,152 | 251,319 | 592,375 |
| D | 43,739 | 187,627 | 166,668 | 398,034 |
| E | 25,768 | 101,419 | 81,397 | 208,583 |
| F | 12,331 | 50,124 | 43,241 | 105,696 |
| G | 15,211 | 52,707 | 78,436 | 146,353 |
| Total | 206,799 | 766,352 | 724,430 | 1,697,580 |

After the base year, the change in the number of new dwellings per archetype drives demand, as previously outlined in section 2.5. All new dwellings mimic the energy intensity of the average base year A-rated dwelling for that archetype. This is due to directive (EU) 2018/844, which requires all new residential dwellings to equate to at least a A2 BER by 2020.

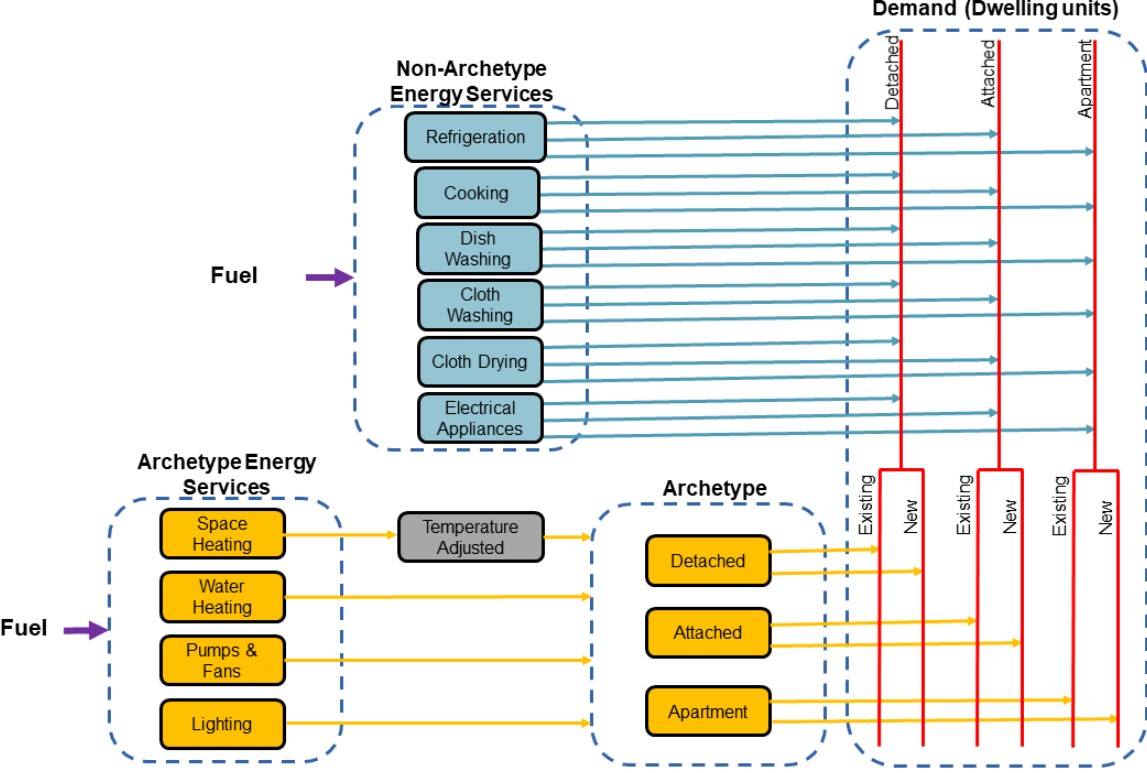

**Figure 5.** Residential Reference Energy System





**Table 9.** Internal Temperature Assumptions

| BER Rating | Living Area Temperature | Non-living Area Temperature |
|:---:|:---:|:---:|
| A | 23°C | 20°C |
| B1 | 21°C | 18°C |
| B2 | 21°C | 18°C |
| B3 | 21°C | 18°C |
| C | 18°C | 15°C |
| D | 18°C | 15°C |
| E | 18°C | 15°C |
| F | 18°C | 15°C |
| G | 18°C | 13°C |

The non-archetype energy service demands are: cooking, refrigeration, cloth washing, cloth drying, dish washing and electrical appliances. The non-archetype energy service demands are not dependent on the age, type or BER rating of the future housing type. These demands are projected to grow at the same rate as the growth in the total housing stock.

    Non-Archetype energy demand data was obtained from SEAI (2018). This data was cross-checked with the JRC-IDEES (Integrated Database of the European Energy System ) database. The process involved cross-checking the share of residential

energy service demands in TIM, against a database which had no input in to the model.

### 3.4.2   Future technology options

The TIM residential sector has two main mitigation options - switching heat technologies and fuels, and retrofitting existing dwellings to improve the thermal efficiency and uplift BER ratings.

    While Ireland has ambitious retrofitting targets for 2030, data on the cost and energy savings for retrofitting is limited,

particularly for deep retrofits. The expected costs and heat energy saving of retrofitting from one BER rating to another requires further investigation. The retrofit cost data is based upon AECOM (2013) and Ali et al. (2020) and the expected energy saving is based upon Collins and Curtis (2017). For this reason, the model uses a simplistic retrofitting options. There are two options for each archetype: shallow retrofit, which reduces dwelling space heating energy demand by 10-34%, and deep retrofit, which improves space heating energy efficiency by at least 35%.

The cost and expected heat energy savings per archetype by BER improvement is outlined in Table 10. The model currently implements a weighted average value, but a fully disaggregated retrofit could be implemented in a future model version.

    The cost and efficiencies of new space heating and water heating technologies is sourced from Danish Energy Agency and Energinet (2020). The cost and efficiency of lighting, pumps & fans and non-archetype demands came from a range of sources including SEAI (2018) and Topten International Group (2021).





**Table 10.** Retrofit Cost & Expected Savings

| BER | Apartment | | Attached | | Detached | |
|---|---|---|---|---|---|---|
| Rating | Cost, EUR | Savings | Cost, EUR | Savings | Cost, EUR | Savings |
| | Deep Retrofit | | | | | |
| C to A | 27,381.6 | 73% | 30,424 | 73% | 33,466.4 | 73% |
| D to B | 11,533.5 | 59% | 12,815 | 60% | 14,096.5 | 60% |
| E to B | 18,273.6 | 70% | 20,304 | 69% | 22,334.4 | 69% |
| F to B | 19,958.4 | 76% | 22,176 | 76% | 24,393.6 | 76% |
| G to B | 19,958.4 | 81% | 22,176 | 81% | 24,393.6 | 81% |
| | Shallow Retrofit | | | | | |
| B to A | 18,957.6 | 55% | 21,064 | 55% | 23,170.4 | 54% |
| C to B | 11,533.5 | 41% | 9,007.5 | 41% | 14,096.5 | 40% |
| D to C | 7,478.1 | 31% | 7,536.3 | 32% | 9,139.9 | 32% |
| E to D | 11,848.5 | 26% | 12,682.5 | 24% | 14,240.8 | 24% |
| F to E | 7,794 | 21% | 8,660 | 22% | 9,526 | 21% |
| G to F | 2,200 | 20% | 2,128 | 20% | 3,200 | 20% |

### 3.4.3 User constraints

– One space and water heating system per household (stoves excluded). This user constraint is applied to represent individual heating systems and calibrate results.

– Maximum 60% of existing buildings to be retrofitted by 2030, and 95% by 2070, the maximum value is interpolated between 2030 and 2050. This user constraint is applied to align with maximum available labour.

– A prerequisite retrofitting requirement for electrical heat pump installation is applied to each archetype. Table 11 shows the percentage of dwellings which requires a shallow or deep retrofit before an electrical heat pump can be installed (A "heat pump ready" dwelling must be at least a B2 energy rating). This user constraint is applied because of the lack of data regrading operation of electrical heat pumps in less than B2 rated dwellings, without this constraint heat pumps would be installed in poorly rated dwellings and run at high performance. The constraint aligns with known heat pump performance data.

**Table 11.** Prerequisite Heat Pump Requirement

| Requirement | Apartment | Attached | Detached |
|---|---|---|---|
| No Retrofit | 24.5% | 14.3% | 12% |
| Shallow Retrofit | 28.5% | 34.7% | 36.8% |
| Deep Retrofit | 47% | 51% | 51.2% |





– Maximum fuel share in cooking for natural gas and LPG is 40% and 10% respectively. Base year fuel share for natural gas is 32% and LPG is 1.5%. This user constraint is applied to align with the increasing market share of electrical induction hobs.

## 3.5 Industry

The industrial sector is modelled using a "top-down" methodology where energy demand is projected based on an assumed future economic growth. Fourteen subsectors are represented and are based on SEAI (2019) energy balance. Baseline shares of energy carriers in the final energy consumption by subsector are assumed constant into the future and are based on the 2018 values (SEAI, 2019).

Energy demand for the industrial sector is projected using GVA per capita for each NACE category and population (Yakut
and de Bruin, 2020). Historical energy consumption is obtained from SEAI (2019) energy balance. The total energy demand from industry in 2050 is projected to increase by 47% from the 2018 level. Cement demand up to 2025 is projected using Department of Finance (2020) stability program update which provides forecasts for the growth in modified investment. In 2019, 65% of the modified investment was in building and construction. Calculating a linear regression of the log of the index for output of the cement sector on the log of investment in building and construction at constant price results in an 18.6%
increase in cement demand in 2025 from 2018 level. Beyond 2025, growth in cement demand is assumed to be the same as the growth in GNI*. This leads to a further increase by 17.8% between 2025 and 2050 at a CAGR of 0.7%. The energy intensity of the industry sector is expected to decline by 46.5% between 2018 and 2050 with a CAGR of -2%, reflecting historical trends.

Fuel switching is the only mitigation option available for combustion emissions from industry in the model. It is controlled through maximum predefined shares which are defined per year and subsector. Table 12 illustrates assumed maximum fuel
switching shares which are possible across all the subsectors.

**Table 12.** Maximum fuel switching shares

| Fuel switching option | 2022 | 2030 | 2050 |
|---|---|---|---|
| Kerosene to biokerosene | 10% | 30% | 100% |
| Diesel to biodiesel | 10% | 30% | 100% |
| Natural gas to biogas | 10% | 30% | 100% |
| Coal and coke to biomass | 8% | 24% | 80% |
| Coal and coke to hydrogen | 2% | 6% | 20% |

### 3.5.1 Process emissions

Among the industrial processes, the cement industry has the largest share of process emissions, which is accounted for in the model. Historical data of cement production and emissions is obtained from the National Inventory Submissions (Duffy et al., 2020). Process emissions increased by 87% between 1990 and 2018 with a CAGR of 2.3%. The cement demand is projected
using the number of new dwellings and the energy consumption of the corresponding industrial sector in a linear model. The





amount of cement needed in 2050 is projected to double relative to 2018 with a CAGR of 2.2%. The demand for cement is then used to project process emission, which is expected to increase by 96% between 2018 and 2050 with a CAGR of 2.1%.

**Table 13.** Process emissions

| Process emissions (kt/PJ) | Reference Case | | with CCS | |
|---|---|---|---|---|
| | 2018 | 2050 | 2018 | 2050 |
| Other non-metallic mineral products demand process | 117 | 155 | 8 | 11 |

## 3.6 Services

The service sector in TIM comprises public and private services. It includes a representation of the following energy ser-
vices: space heating, space cooling, water heating, cooking, refrigeration, building lighting and other appliances. Data centres electricity demand and public lighting are also represented.

Future fuel switching and technology-based efficiency options in the Services sector are represented explicitly. However given a lack of sufficient building-level data to enable a detailed analysis, public and private services are modelled in an aggregated fashion (i.e. the building stock is not divided in categories). This is an area identified as a priority area for future
model development.

The end-use demand services in the service sector include million meter square of area for public and private sector buildings, data centre demand and public lighting units. The area is projected assuming the same growth as in GNI*. The total area is projected to increase by 61.8% by 2050 from 2018 level with a CAGR of 1.5%.

Electricity demand for data centres is obtained from EirGrid's "steady evolution" scenario (EirGrid, 2017). The demand
is expected to increase by 6 times in 2030 from 2018 level. We assume no growth in data centre demand after 2030 since permission requests for new/expanding existing capacities are not available yet. Further, post-2030 demand is even more uncertain given (i) the potential for exponential data usage growth and further data centres applications vs (ii) technology improvements and whether data centres fully utilise their contracted import capacity. Electricity consumption of data centres for cooling, and their potential to supply district heat from the excess heat they generate, are represented using the methodology
described in Petrović et al. (2020).

Public lighting units are projected based on the Project Ireland 2040, whereby five major cities of Ireland, Dublin, Cork, Limerick, Galway and Waterford are expected to grow by 50% in 2040 (Government of Ireland, 2018). This results in a 12.5% increase in public lighting units in Ireland by 2040 from 2018 level, with a CAGR of 1%. Beyond 2040, the units are projected to increase by 1% per annum until 2050.

## 3.7 Agriculture

The current version of the agriculture sector in TIM comes from the Irish TIMES model and is documented in Chiodi et al. (2016). It includes representation of 12 energy service demands; half of them belong to the livestock and half to the tillage





sector. Land availability and water consumption are explicitly represented and accounted for in the sector, however no specific constraints are set. Future energy service demands in the agriculture sector are assumed to be unchanged from the 2018 level.

## 4  Discussion & conclusion

This section discusses the strengths, limitations and development priorities for the TIMES-Ireland model, using the framework proposed by Pye et al. (2021) and DeCarolis et al. (2017), which outline best practice for ESOM development and priorities for new developments and applications of ESOMs for deep decarbonisation challenges.

### 4.1  Analytical advancements

The following is a summary of the main analytical advancements of TIM, explained in more detail throughout the text.

- The model has a flexible time-slice configuration with resolution possible up to the hourly level, which is necessary to e.g. model the power system under very high shares of variable renewables in deep decarbonisation scenarios, including power storage.

- The model also makes use of new TIMES features in the power sector, modelling dispatch, unit commitment and capacity expansion.

- The model has been developed with flexible regional definitions, with transport and power sectors detailed at the county level.

- Energy service demands are driven by a consistent set of population and macro-economic indicators, consistent with a national CGE model, and also disaggregated in sufficient detail to allow alternative demand scenarios, such as transport mode shifting, lower housing demand requirements etc.

- The development process strives to achieve best practice in software development, including version control, transparency (with open source and open data), quality assurance, documentation and wide stakeholder consultation.

### 4.2  Modelling for policy insight

The main purpose of the model is to meet the policy need to inform detailed sectoral pathways which can meet very ambitious decarbonisation targets. This subsection discusses TIM's strengths and weaknesses in this regard.

Firstly, the integrated whole system approach offer "macro systems" perspective. The temporal and spatial flexibility resolves the tension between the requirements for granularity for certain model functionalities, and the requirement for data and model tractability, and fast solving speed. Integration with key national data sources and other models is a further strength of TIM.

Secondly, an appropriately wide mitigation option space is required to meet deep decarbonisation challenges. Strengths of TIM include the rich depiction of important demand sectors, namely transport and residential, which are characterised to enable





scenario variants of alternative demands. For example, splitting passenger transport demand into long-, medium- and short-distance demand allows for switching to active and public transport demands . Similarly, the archetypal model underpinning the residential sector allows for lowering building floor area demand and internal temperatures, which can be key scenario variants. These scenario variants are exogenous, though there is increasing literature on endogenising behaviour in TIMES

models. Several options for shifting to more efficient uses of energy, for example, are endogenous. However, the aggregate nature of the Industry sector limits the potential analysis of demand reduction in this sector and is an area of future model development priority.

Another area for future model improvements is to explore carbon dioxide removal (CDR) options in more detail. At the global level, modelled pathways meeting the Paris Agreement rely on large-scale deployment of CDR options including

biomass with carbon capture and storage (BECCS) (IPCC, 2018). BECCS is currently modelled within TIM, but options like Direct Air Capture (DAC) (Realmonte et al., 2019) are currently represented in an oversimplified manner; as real-world trials provide operational performance data, we will be able to improve the representation of DAC in TIM.

Another limitation of TIM is the sole focus on energy and process emissions. Agriculture and land-use are very significant emitters in Ireland, and future energy system decarbonisation trajectories will require a focus on the overlaps with other

emitters. This is required firstly to understand bioenergy and waste potentials, and to take into account future gains from the circular economy, and also competition between energy crops, agriculture and reforestation.

A focus on the practice of model development and application is key. Robust quantitative analysis and information is an important ingredient in the energy and climate policy making process. But energy modelling should support the policy making process rather than determine its contents. This means the interface between energy modelling and the policy making process

should be an iterative one that incorporates regular review and feedback as new issues and questions emerge (Strachan et al., 2016). TIM has been developed with the aims of openness and transparency, quality assurance and best practice in software development, and iterative engagement with key stakeholders to feed in sectoral and industry expertise, and to be able reflect policy and societal factors. Any model built to inform policy should be open to deep scrutiny early and throughout the process. It also requires capacity-development among the "consumers" of model outputs to ensure that results are interpreted,

communicated and applied appropriately.

It is also important to communicate what the model can and can't do. For example, while TIM is a "cost-optimal" model, it does not capture many of the societal costs or benefits that the energy system imposes. For example, higher renewable energy shares may represent a higher overall investment and operation cost over the project lifetime, these may offer significant societal benefits in terms of lower air pollution, reducing the fossil fuel import bill and providing high-quality employment. Similarly,

the distributional effects of the costs are not captured in the model, therefore considerations of a "just transition" are a challenge within TIM currently.

### 4.3   Future improvements

TIM will undergo continuous and iterative improvements and developments. In the near-term, the following is the priority list for future development.





- Modelling gas supply at hourly resolution - currently modelled at a seasonal level, which may unintentionally allow for energy storage capacity.

- Similarly, electricity inter-connectors with the UK and France can be modelled at the same time resolution as the power sector, taking into account capacity constraints on top of annual trade constraints.

- Modelling hydrogen production routes other than from renewables (green hydrogen), such as blue, brown and grey hydrogen production options.

- A review and update of bioenergy conversion options, including an update of domestic bioenergy potentials.

- A review and update of future low-carbon technology costs.

- Developing further the Agri-TIMES module, first developed by Chiodi et al. (2016), in order to model agriculture, livestock and land-use emissions and mitigation options explicitly, including competition for land-use.

- A more detailed bottom-up focus on the Industry sector, which is currently modelled in an aggregated top-down fashion, and disaggregation of the Services sector.

- Developing a "Business as usual"/"With existing policy measures" case, which includes current policies and measures and hurdle rates to represent end-use technology uptake.

## 5 Conclusions

This paper presents the TIMES-Ireland Model, an energy systems optimisation model which is a significant step forward both in terms of analytical heft, open and best-practice development approach and in its contribution to national evidence-based policy development. TIM has been developed to help inform very ambitious decarbonisation objectives and to inform future carbon budgets, on the path to net-zero GHG emissions by 2050.

The model was developed from the legacy model, Irish TIMES, which has had a long history of contributing the the evidence base for Irish energy policy and for pushing the state-of-the-art in ESOM development. TIM also benefits from ongoing collaborations and interactions with other national models and data sources, including LEAP-Ireland, the ESRI's I3E model and SEAI databases. The model also benefits from the continuous development of the TIMES model generator within the ETSAP community.

Continuous, iterative and open model development is essential to ensure that the model remains fit-for-purpose and state-of-the-art. The model has been built with these future calibrations and improvements in mind, with clear open documentation and development protocol to allow for ongoing improvements and updates, to better enable the Irish energy system to achieve the goal of carbon neutrality.





*Code and data availability.* The version of the TIMES source code used to develop and execute TIMES-Ireland Model is available on Zenodo: https://doi.org/10.5281/zenodo.5537496. The TIMES-Ireland Model (which is essentially a database composed of excel files) is available on GitHub (https://github.com/MaREI-EPMG/times-ireland-model) and is archived on Zenodo (https://doi.org/10.5281/zenodo.5708680).





## Appendix A: Supplementary techno-economic assumptions

**Table A1.** Import Fossil fuel commodity prices

| Technology Description | 2018 | 2019 | 2025 | 2030 | 2035 | 2040 | *Source/Ratio |
|---|---|---|---|---|---|---|---|
| | EUR/GJ | EUR/GJ | EUR/GJ | EUR/GJ | EUR/GJ | EUR/GJ | |
| Crude Oil | 10.1 | 9.9 | 11.1 | 11.9 | 12.7 | 13.3 | WEO2019/2020 |
| Natural Gas - UK | 6.1 | 5.7 | 6.2 | 9.4 | 11.0 | 12.9 | WEO2019/2020 |
| Hard Coal / Antracite | 1.9 | 1.3 | 1.4 | 1.7 | 1.9 | 2.1 | WEO2019/2020 |
| Bituminous Coal | 1.8 | 1.2 | 1.3 | 1.6 | 1.8 | 2.0 | 0.95 |
| Coke Coal | 2.4 | 1.7 | 1.8 | 2.2 | 2.4 | 2.6 | 1.27 |
| Lignite / Brown Coal | 1.6 | 1.1 | 1.2 | 1.5 | 1.6 | 1.8 | 0.88 |
| Liquefied Natural Gas | 6.5 | 6.1 | 6.6 | 10.0 | 11.7 | 13.8 | 1.07 |
| Diesel Oil | 15.4 | 15.1 | 17.0 | 18.2 | 19.4 | 20.3 | 1.53 |
| Gasoline | 16.6 | 16.2 | 18.3 | 19.6 | 20.9 | 21.9 | 1.65 |
| Heavy Fuel Oil | 8.2 | 8.0 | 9.0 | 9.6 | 10.3 | 10.8 | 0.81 |
| Kerosene | 16.6 | 16.2 | 18.3 | 19.6 | 20.9 | 21.9 | 1.65 |
| Liquefied Petroleum Gas | 13.0 | 12.8 | 14.4 | 15.4 | 16.4 | 17.2 | 1.29 |
| Petroleum Coke | 16.6 | 16.2 | 18.3 | 19.6 | 20.9 | 21.9 | 1.65 |
| Uranium | | | | | | | |
| Oil for Non-Energy uses | 10.1 | 9.9 | 11.1 | 11.9 | 12.7 | 13.3 | |

**Table A2.** Import bioenergy commodity prices

| Bioenergy Import Costs | 2015 | 2020 | 2025 | 2030 | 2040 | 2050 |
|---|---|---|---|---|---|---|
| Technology Description | EUR/GJ | EUR/GJ | EUR/GJ | EUR/GJ | EUR/GJ | EUR/GJ |
| Import of Ethanol 1st generation - Step 1 | 18.18 | 17.58 | 16.60 | 15.88 | | |
| Import of Ethanol 1st generation - Step 2 | 18.92 | 19.08 | 18.92 | 19.13 | 20.39 | 21.40 |
| Import of Ethanol 1st generation - Step 3 | 19.68 | 20.68 | 21.76 | 23.55 | 25.10 | 26.34 |
| Import of Ethanol 1st generation - Step 4 | 20.64 | 22.71 | 25.17 | 28.76 | 30.65 | 32.16 |
| Import of Biodiesel 1st generation - Step 1 | 31.36 | 33.18 | 31.86 | 31.34 | | |
| Import of Biodiesel 1st generation - Step 2 | 32.65 | 35.83 | 35.97 | 37.02 | 39.46 | 41.41 |
| Import of Biodiesel 1st generation - Step 3 | 33.92 | 38.33 | 40.46 | 43.76 | 46.64 | 48.94 |
| Import of Biodiesel 1st generation - Step 4 | 35.54 | 41.92 | 46.53 | 52.83 | 56.31 | 59.09 |
| Import of Wood Pellets - Step 1 | 10.65 | 8.50 | 7.28 | 6.95 | | |
| Import of Wood Pellets - Step 2 | 11.03 | 9.05 | 8.07 | 7.91 | | |
| Import of Wood Pellets - Step 3 | 12.28 | 10.10 | 8.93 | 8.77 | | |
| Import of Wood Pellets - Step 4 | 12.78 | 10.80 | 9.98 | 10.06 | | |
| Import of Wood Chip - Step 1 | 5.21 | 4.16 | 3.56 | 3.39 | | |
| Import of Wood Chip - Step 2 | 5.40 | 4.42 | 3.94 | 3.87 | | |
| Import of Wood Chip - Step 3 | 6.14 | 5.04 | 4.47 | 4.37 | | |
| Import of Wood Chip - Step 4 | 6.38 | 5.40 | 4.99 | 5.04 | | |





**Table A3.** Import bioenergy delivery costs

| Bioenergy Import Delivery Costs Technology Description | 2020 EUR/GJ | 2030 EUR/GJ | 2040 EUR/GJ | 2050 EUR/GJ | Notes |
|---|---|---|---|---|---|
| Import of Wood Pellets - Step 1 | 4.32 | 4.45 | 4.59 | 4.69 | |
| Import of Wood Pellets - Step 2 | 4.32 | 4.45 | 4.59 | 4.69 | |
| Import of Wood Pellets - Step 3 | 4.32 | 4.45 | 4.59 | 4.69 | |
| Import of Wood Pellets - Step 4 | 4.32 | 4.45 | 4.59 | 4.69 | |
| Import of Wood Chip - Step 1 | 4.32 | 4.45 | 4.59 | 4.69 | |
| Import of Wood Chip - Step 2 | 4.32 | 4.45 | 4.59 | 4.69 | |
| Import of Wood Chip - Step 3 | 4.32 | 4.45 | 4.59 | 4.69 | |
| Import of Wood Chip - Step 4 | 4.32 | 4.45 | 4.59 | 4.69 | |

**Table A4.** Import bioenergy potentials

| Imported Bioenergy potentials (PJ) Technology Description | 2018 PJ | 2020 PJ | 2025 PJ | 2030 PJ | 2035 PJ | 2040 PJ | 2045 PJ | 2050 PJ |
|---|---|---|---|---|---|---|---|---|
| Import of Ethanol 1st generation - Step 1 | 1.0 | 1.0 | 1.0 | 1.0 | 1.0 | 1.0 | 1.0 | 1.0 |
| Import of Ethanol 1st generation - Step 2 | 0.0 | 1.9 | 1.9 | 1.9 | 1.9 | 1.9 | 1.9 | 1.9 |
| Import of Ethanol 1st generation - Step 3 | 0.0 | 3.8 | 3.8 | 3.8 | 3.8 | 3.8 | 3.8 | 3.8 |
| Import of Ethanol 1st generation - Step 4 | 0.0 | 27.1 | 27.1 | 27.1 | 27.1 | 27.1 | 27.1 | 27.1 |
| | | | | | | | | |
| Import of Biodiesel 1st generation - Step 1 | 4.4 | 4.4 | 4.4 | 4.4 | 4.4 | 4.4 | 4.4 | 4.4 |
| Import of Biodiesel 1st generation - Step 2 | 0.0 | 8.7 | 8.7 | 8.7 | 8.7 | 8.7 | 8.7 | 8.7 |
| Import of Biodiesel 1st generation - Step 3 | 0.0 | 17.4 | 17.4 | 17.4 | 17.4 | 17.4 | 17.4 | 17.4 |
| Import of Biodiesel 1st generation - Step 4 | 0.0 | 72.2 | 72.2 | 72.2 | 72.2 | 72.2 | 72.2 | 72.2 |
| | | | | | | | | |
| Import of Wood Pellets - Step 1 | 0.7 | 0.7 | 0.7 | 0.7 | 0.7 | 0.7 | 0.7 | 0.7 |
| Import of Wood Pellets - Step 2 | 0.0 | 1.4 | 1.4 | 1.4 | 1.4 | 1.4 | 1.4 | 1.4 |
| Import of Wood Pellets - Step 3 | 0.0 | 2.7 | 2.7 | 2.7 | 2.7 | 2.7 | 2.7 | 2.7 |
| Import of Wood Pellets - Step 4 | 0.0 | 105.4 | 105.4 | 105.4 | 105.4 | 105.4 | 105.4 | 105.4 |
| | | | | | | | | |
| Import of Wood Chip - Step 1 | 0.3 | 0.3 | 0.3 | 0.3 | 0.3 | 0.3 | 0.3 | 0.3 |
| Import of Wood Chip - Step 2 | 0.0 | 0.6 | 0.6 | 0.6 | 0.6 | 0.6 | 0.6 | 0.6 |
| Import of Wood Chip - Step 3 | 0.0 | 1.2 | 1.2 | 1.2 | 1.2 | 1.2 | 1.2 | 1.2 |
| Import of Wood Chip - Step 4 | 0.0 | 34.7 | 34.7 | 34.7 | 34.7 | 34.7 | 34.7 | 34.7 |



Table A5: Base year generation units

| Technology description Units | Efficiency | Availability | Life Years | Start Year | Capacity GW |
|---|---|---|---|---|---|
| PWR Renewable: Hydropower dam and reservoir HYD Existing Ardnacrusha 1 | 1.00 | 0.45 | 150 | 1929 | 0.02 |
| PWR Renewable: Hydropower dam and reservoir HYD Existing Ardnacrusha 2 | 1.00 | 0.45 | 150 | 1929 | 0.02 |
| PWR Renewable: Hydropower dam and reservoir HYD Existing Ardnacrusha 3 | 1.00 | 0.44 | 150 | 1929 | 0.02 |
| PWR Renewable: Hydropower dam and reservoir HYD Existing Ardnacrusha 4 | 1.00 | 0.45 | 150 | 1929 | 0.02 |
| PWR Thermal Power Plant: Combined cycle gas turbine GAS Existing Aghada CCGT | 0.70 | 0.90 | 30 | 2010 | 0.43 |
| PWR Thermal Power Plant: Open cycle gas turbine GAS Existing Aghada CT 1 | 0.42 | 0.95 | 43 | 1980 | 0.09 |
| PWR Thermal Power Plant: Open cycle gas turbine GAS Existing Aghada CT 2 | 0.42 | 0.95 | 44 | 1980 | 0.09 |
| PWR Thermal Power Plant: Open cycle gas turbine GAS Existing Aghada CT 4 | 0.42 | 0.95 | 44 | 1980 | 0.09 |
| PWR Thermal Power Plant: Combined cycle gas turbine GAS Existing Ballylumford CCGT Unit 10 | 0.60 | 0.90 | 35 | 2003 | 0.10 |
| PWR Thermal Power Plant: Combined cycle gas turbine GAS Existing Ballylumford CCGT Unit 31 | 0.63 | 0.90 | 35 | 2003 | 0.25 |
| PWR Thermal Power Plant: Combined cycle gas turbine GAS Existing Ballylumford CCGT Unit 32 | 0.63 | 0.90 | 35 | 2003 | 0.25 |
| PWR Thermal Power Plant: Open cycle gas turbine DIS Existing Ballylumford GT1 | 0.33 | 0.95 | 50 | 1976 | 0.06 |
| PWR Thermal Power Plant: Open cycle gas turbine DIS Existing Ballylumford GT2 | 0.33 | 0.95 | 50 | 1976 | 0.06 |
| PWR Thermal Power Plant: Open cycle gas turbine DIS Existing Coolkeeragh OCGT | 0.33 | 0.95 | 35 | 2005 | 0.05 |
| PWR Thermal Power Plant: Open cycle gas turbine GAS Existing Contour Global Agg Unit | 0.45 | 0.95 | 35 | 2010 | 0.01 |
| PWR Thermal Power Plant: Combined cycle gas turbine GAS Existing Coolkeeragh CCGT | 0.68 | 0.90 | 35 | 2005 | 0.41 |
| PWR Thermal Power Plant: Combined cycle gas turbine GAS Existing Dublin Bay CCGT | 0.68 | 0.90 | 30 | 2002 | 0.41 |
| PWR Thermal Power Plant: Steam turbine PEA Existing Edenderry | 0.49 | 0.90 | 23 | 2000 | 0.12 |
| PWR Thermal Power Plant: Open cycle gas turbine DIS Existing Cushaling | 0.38 | 0.95 | 30 | 2010 | 0.06 |
| PWR Thermal Power Plant: Open cycle gas turbine DIS Existing Cushaling | 0.38 | 0.95 | 30 | 2010 | 0.06 |





**Table A5 – continued from previous page**

| Technology description Units | Efficiency | Availability | Life Years | Start Year | Capacity GW |
|---|---|---|---|---|---|
| PWR Renewable: Hydropower dam and reservoir HYD Existing Erne 1 | 1.00 | 0.47 | 150 | 1950 | 0.01 |
| PWR Renewable: Hydropower dam and reservoir HYD Existing Erne 2 | 1.00 | 0.43 | 150 | 1950 | 0.01 |
| PWR Renewable: Hydropower dam and reservoir HYD Existing Erne 3 | 1.00 | 0.47 | 150 | 1950 | 0.02 |
| PWR Renewable: Hydropower dam and reservoir HYD Existing Erne 4 | 1.00 | 0.40 | 150 | 1950 | 0.02 |
| PWR Thermal Power Plant: Combined cycle gas turbine GAS Existing Great Island CCGT | 0.71 | 0.90 | 35 | 2014 | 0.43 |
| PWR Thermal Power Plant: Combined cycle gas turbine GAS Existing Huntstown Phase II | 0.75 | 0.90 | 30 | 2007 | 0.41 |
| PWR Thermal Power Plant: Combined cycle gas turbine GAS Existing Huntstown | 0.70 | 0.90 | 30 | 2002 | 0.34 |
| PWR Thermal Power Plant: Combined cycle gas turbine DIS Existing iPower AGU | 0.34 | 0.90 | 35 | 2011 | 0.00 |
| PWR Thermal Power Plant: Steam turbine COA Existing Kilroot Unit 1 FGD | 0.18 | 0.90 | 50 | 1981 | 0.20 |
| PWR Thermal Power Plant: Steam turbine COA Existing Kilroot Unit 2 FGD | 0.18 | 0.90 | 50 | 1982 | 0.20 |
| PWR Thermal Power Plant: Open cycle gas turbine DIS Existing Kilroot GT1 | 0.33 | 0.95 | 35 | 2009 | 0.03 |
| PWR Thermal Power Plant: Open cycle gas turbine DIS Existing Kilroot GT2 | 0.33 | 0.95 | 35 | 2009 | 0.03 |
| PWR Thermal Power Plant: Open cycle gas turbine DIS Existing Kilroot GT3 | 0.39 | 0.95 | 35 | 2009 | 0.04 |
| PWR Thermal Power Plant: Open cycle gas turbine DIS Existing Kilroot GT4 | 0.39 | 0.95 | 35 | 2009 | 0.04 |
| PWR Renewable: Hydropower dam and reservoir HYD Existing Lee 1 | 1.00 | 0.33 | 150 | 1957 | 0.02 |
| PWR Renewable: Hydropower dam and reservoir HYD Existing Lee 2 | 1.00 | 0.33 | 150 | 1957 | 0.00 |
| PWR Renewable: Hydropower dam and reservoir HYD Existing Lee 3 | 1.00 | 0.33 | 150 | 1957 | 0.01 |
| PWR Renewable: Hydropower dam and reservoir HYD Existing Liffey 1 | 1.00 | 0.11 | 150 | 1938 | 0.02 |
| PWR Renewable: Hydropower dam and reservoir HYD Existing Liffey 2 | 1.00 | 0.10 | 150 | 1938 | 0.02 |
| PWR Renewable: Hydropower dam and reservoir HYD Existing Liffey 4 | 1.00 | 0.11 | 150 | 1938 | 0.00 |





**Table A5 – continued from previous page**

| Technology description Units | Efficiency | Availability | Life Years | Start Year | Capacity GW |
|---|---|---|---|---|---|
| PWR Renewable: Hydropower dam and reservoir HYD Existing Liffey 5 | 1.00 | 0.11 | 150 | 1938 | 0.00 |
| PWR Thermal Power Plant: Steam turbine PEA Existing Lough Ree | 0.41 | 0.90 | 16 | 2004 | 0.09 |
| PWR Thermal Power Plant: Steam turbine COA Existing Moneypoint 1 | 0.34 | 0.90 | 38 | 1987 | 0.29 |
| PWR Thermal Power Plant: Steam turbine COA Existing Moneypoint 2 | 0.34 | 0.90 | 38 | 1987 | 0.29 |
| PWR Thermal Power Plant: Steam turbine COA Existing Moneypoint 3 | 0.34 | 0.90 | 38 | 1987 | 0.29 |
| PWR Thermal Power Plant: Open cycle gas turbine GAS Existing North Wall 5 | 0.41 | 0.95 | 30 | 2012 | 0.10 |
| PWR Thermal Power Plant: Combined cycle gas turbine GAS Existing Poolbeg C_A | 0.58 | 0.90 | 30 | 2000 | 0.26 |
| PWR Thermal Power Plant: Combined cycle gas turbine GAS Existing Poolbeg C_B | 0.58 | 0.90 | 30 | 2000 | 0.26 |
| PWR Thermal Power Plant: Open cycle gas turbine DIS Existing Rhode 1 | 0.37 | 0.95 | 30 | 2004 | 0.05 |
| PWR Thermal Power Plant: Open cycle gas turbine DIS Existing Rhode 2 | 0.37 | 0.95 | 30 | 2004 | 0.05 |
| PWR Thermal Power Plant: Open cycle gas turbine GAS Existing Sealrock 3 (Aughinish CHP) | 0.72 | 0.93 | 35 | 2005 | 0.08 |
| PWR Thermal Power Plant: Open cycle gas turbine GAS Existing Sealrock 4 (Aughinish CHP) | 0.72 | 0.93 | 35 | 2005 | 0.08 |
| PWR Thermal Power Plant: Steam turbine HFO Existing Tarbert Unit 1 | 0.31 | 0.95 | 54 | 1969 | 0.05 |
| PWR Thermal Power Plant: Steam turbine HFO Existing Tarbert Unit 2 | 0.31 | 0.95 | 54 | 1969 | 0.05 |
| PWR Thermal Power Plant: Steam turbine HFO Existing Tarbert Unit 3 | 0.42 | 0.95 | 54 | 1969 | 0.24 |
| PWR Thermal Power Plant: Steam turbine HFO Existing Tarbert Unit 4 | 0.40 | 0.95 | 54 | 1969 | 0.24 |
| PWR Thermal Power Plant: Open cycle gas turbine DIS Existing Tawnaghmore 1 | 0.38 | 0.95 | 30 | 2003 | 0.05 |
| PWR Thermal Power Plant: Open cycle gas turbine DIS Existing Tawnaghmore 3 | 0.38 | 0.95 | 30 | 2003 | 0.05 |
| PWR Thermal Power Plant: Combined cycle gas turbine GAS Existing Tynagh | 0.68 | 0.90 | 30 | 2006 | 0.40 |
| PWR Thermal Power Plant: Combined cycle gas turbine GAS Existing Whitegate | 0.75 | 0.90 | 30 | 2010 | 0.45 |

| Table A5 – continued from previous page | | | | | |
|---|---|---|---|---|---|
| Technology description Units | Efficiency | Availability | Life Years | Start Year | Capacity GW |
| PWR Thermal Power Plant: Steam turbine PEA Existing West Offaly | 0.42 | 0.90 | 15 | 2005 | 0.14 |
| PWR Storage: Pumped Storage HYD Existing Turlough Hill 1 | 0.75 | 0.16 | 150 | | 0.07 |
| PWR Storage: Pumped Storage HYD Existing Turlough Hill 2 | 0.75 | 0.16 | 150 | | 0.07 |
| PWR Storage: Pumped Storage HYD Existing Turlough Hill 3 | 0.75 | 0.16 | 150 | | 0.07 |
| PWR Storage: Pumped Storage HYD Existing Turlough Hill 4 | 0.75 | 0.16 | 150 | | 0.07 |

**Table A6.** Residential Heating Cost & Efficiency. Note: it is assumed a water heaters are 30% less efficient than space heaters. The costs shown are for space heaters only.

| Description | | 2020 | | 2030 | | 2040 | | 2050 | |
|---|---|---|---|---|---|---|---|---|---|
| Archetype | Technology | kEUR | $\eta$ | kEUR | $\eta$ | kEUR | $\eta$ | kEUR | $\eta$ |
| | Gas Boiler | 2.79 | 81% | 2.79 | 81% | 2.79 | 81% | 2.79 | 81% |
| Apartment | Oil Boiler | 3.63 | 81% | 3.63 | 81% | 3.63 | 81% | 3.63 | 81% |
| | Heat Pump | 7.5 | 307% | 6.83 | 338% | 6.21 | 379% | 6.15 | 409% |
| | Gas Boiler | 3.25 | 82% | 3.25 | 82% | 3.25 | 82% | 3.25 | 82% |
| Attached | Oil Boiler | 4.23 | 82% | 4.23 | 82% | 4.23 | 82% | 4.23 | 82% |
| | Heat Pump | 8.53 | 312% | 7.76 | 343% | 7.06 | 384% | 6.99 | 415% |
| | Gas Boiler | 3.53 | 83% | 3.53 | 83% | 3.53 | 83% | 3.53 | 83% |
| Detached | Oil Boiler | 4.58 | 82% | 4.58 | 82% | 4.58 | 82% | 4.58 | 82% |
| | Heat Pump | 9.85 | 309% | 8.96 | 340% | 8.15 | 381% | 8.07 | 412% |



**Table A7.** Techno-economic characteristics of passenger transport vehicles

| Technology (fuel) | Fuel economy (million vkm/PJ) | | Purchase price[a] (€2018) | | Annual maintenance cost (€2018) |
|---|---|---|---|---|---|
| | 2018 | 2050 | 2018 | 2050 | |
| **Light-duty vehicles** | | | | | |
| ICE (Gasoline) | 413 | 413 | 20,290 | 20,290 | 1,015 |
| ICE (E85) | 389 | 418 | 20,290 | 20,290 | 1,015 |
| ICE (Diesel/B20) | 573 | 615 | 21,832 | 21,832 | 1,092 |
| ICE (B100) | 556 | 596 | 21,832 | 21,832 | 1,092 |
| ICE (Dual-fuel) | 413 | 413 | 20,290 | 20,290 | 1,015 |
| ICE (CNG/BioCNG) | 413 | 443 | 24,631 | 24,631 | 1,232 |
| HEV (Gasoline) | 556 | 596 | 23,752 | 22,769 | 1,188 |
| HEV (Diesel) | 719 | 772 | 25,646 | 24,569 | 1,282 |
| PHEV (Gasolone, Electricity) | 940 | 1,081 | 30,950 | 25,371 | 1,547 |
| PHEV (Diesel, Electricity) | 1,216 | 1,398 | 33,424 | 27,377 | 1,671 |
| BEV (Electricity) | 1,623 | 1,886 | 32,971 | 24,646 | 1,649 |
| FCV (Hydrogen) | 882 | 1,012 | 60,819 | 24,796 | 3,041 |
| **Buses** | | | | | |
| ICE (Diesel) | 106 | 114 | 109,959 | 113,565 | 5,498 |
| ICE (B100) | 106 | 114 | 109,959 | 113,565 | 5,498 |
| ICE (CNG/BioCNG) | 101 | 108 | 109,959 | 113,565 | 5,498 |
| BEV (electricity) | 337 | 391 | 397,219 | 130,000 | 19,861 |
| FCV (hydrogen) | 192 | 236 | 397,219 | 130,689 | 19,861 |
| **Train** | | | | | |
| Light train BEV (Electricity) | 21 | 21 | 231,583 | 231,583 | 11,579 |
| Heavy train BEV (Electricity) | 21 | 21 | 935,522 | 935,522 | 46,776 |
| Heavy train ICE (Diesel) | 8 | 8 | 989,272 | 989,272 | 49,464 |

[a] The values exclude taxes and subsidies





*Author contributions.* OB conceptualised the idea together with HD and JG. HD led preparation of the manuscript with contributions from JG, VA, AG, JM, XY and OB. All authors contributed to model creation, data analysis and model validation. OB and AS reviewed and edited
the manuscript before the original submission.

*Competing interests.* The authors declare that they have no conflict of interest.

*Acknowledgements.* This work is being undertaken as part of a number of research projects undertaken at the SFI MaREI Centre at University College Cork, including CAPACITY, funded by the Department of Environment, Climate and Communications and the SFI-funded CHIMERA project[4].
The authors wish to acknowledge the contribution of the following people: Alessandro Chiodi, Paul Deane, Maurizio Gargiulo, Gerald Lyons, Brian Ó Gallachóir, Fionn Rogan.
    The authors would also like to thank the following experts for their feedback during the expert review process: Andrew Kelly, Sarah Ni Ruairc, Joseph Cummins, Andrew Moloney, Marie Bourke, Ilkka Keppo, John Curtis, Gerry Duggan, John Fitzgerald, Luke O'Call White, Tadhg O'Mahony, Andrew Murphy, as well as individuals at UCD, DFIN, and EirGrid.

---

[4]CHIMERA is supported by a research grant from Science Foundation Ireland (SFI) and the National Natural Science Foundation of China (NSFC) under the SFI-NSFC Partnership Programme, grant no. 17/NSFC/5181 and supported by MaREI, the SFI Research Centre for Energy, Climate, and Marine [Grant No: 12/RC/2302_P2], https://www.marei.ie/project/chimera/





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
