# Peer review of "TIM: Modelling pathways to meet Ireland's long-term energy system challenges with the TIMES-Ireland Model (v1.0)"

_Geoscientific Model Development, 2021_

## Referee Comment (RC2)

[referee-annotated manuscript omitted]

---

## Author Response (AR1)

**Author's response**

Dear Editor,

In the following we provide a point-by-point response to the points raised by the reviewers.

We would like to once again thank both reviewers for a detailed review of the manuscript and their constructive comments.

Best regards,

Olexandr Balyk

**Reviewer 1**

This paper describes a new, state-of-the-art energy systems optimisation model for Ireland which has been developed following best practice guidelines. The objective of the model is to support evidenced based policy making as Ireland seeks to reach net-zero by 2050.

In general I think the paper, and the model it describes, are of a high quality and will certainly warrant publication once a few minor points below are clarified and amended.

1. Interesting choice not to include technology specific discount rates – is there any further discussion/rationale you could give on this issue? That is, I understand there is a tension here given that the model takes a social planner perspective but I am wondering about this choice since technology specific discount rates are typically used to, in part, acknowledge that the real world generally relies on private finance (at least as of now).

*It's a simplifying assumption used to reflect a highly uncertain future. We acknowledge that real-world deployment, relies - and is likely to continue to rely for the immediate future - on private finance for some (but not all) of the decarbonisation transition. However, the immediate use of this model has been for Ireland's planning of carbon budgets under a very tight 2030 target; these scenarios all involve significant expenditure in the period 2022-2030 - particularly for household items (EVs, zero-carbon heating, insulation) - and it is very likely that substantial public sector funding over that period will be required to deliver the budgets and targets. As such, for this exercise, a tech-agnostic single discount rate has been used. We do not rule out the use of tech-specific discount rates in future, particularly for other scenarios where direct private-sector funding dominates the transition.*

2. I'm a little unclear if land-use emissions are modelled in TIM based on "Another limitation of TIM is the sole focus on energy and process emissions". My impression is they are but in a static, exogenous manner? Even though there is mention of the agricultural sector being taken from Irish TIMES, it would be useful to clarify, with a few sentences, what is exogenous and endogenous here.

*Thank you for pointing this out. We have expanded the description of the agricultural sector to clarify this and other assumptions.*

3. Line 20 – I think this should be updated to "did" fail its 2020 decarbonisation objective, as it is now 2022.

*Changed to "was expected to fail", based on the doc cited which refers to estimated rather than actual data.*

4. Line 36 – make improving efficiencies challenging (previously "make improve..")

*Corrected, thanks you!*

5. Line 37 – is that average annual renewable electricity generation (36.5%?) And again, is 86% average from wind. It will vary between years.

*Clarified this by changing to: "… 42% of its electricity from renewables in 2020, 86% of which came from wind energy…".*

6. Line 76 – the model respects whatever tech, environ, economic, social and policy constraints are included – some may be missed (and likely are).

*We agree that although the model respects whatever tech, environmental, economic and policy constraints are included, in real life policy targets, environmental goals, etc may be missed.*

7. Line 78 – how are feasible uptake rates derived?

*These tend to be scenario specific and can be based on e.g. historical trends or expert judgement.*

8. Line 92 – what is an "internal transfer"?

*Changed the respective sentence to the following to clarify this: "Profits, taxes and subsidies are internal transfers, i.e. occurring within the economy, that do not change the NPV (albeit taxes and subsidies can be included to influence the optimisation)."*

9. Figure 1 – I'm going to assume it can but from the diagram it isn't totally clear that electricity can be used to produce H2, for instance. And is there any CCU?

*Thank you for spotting this! The figure has been modified to clarify that electricity can be used to produce H2. No CCU is available in this model version.*

10. Line 143 – what is a "spacial spatio-temporal approach"?

*Corrected the typo, thank you!*

11. Line 260 – can interconnectors be reinforced/new ones built?

*Added the following sentences to clarify this point:*

- *"Greenlink and Celtic Interconnector are assumed operational from 2023 and 2026 respectively."*
- *"Neither reinforcement of existing nor investment in new interconnectors are currently included in the model".*

12. Line 272 – I think the sentence should be something like "…in the future, disaggregated as (or into) centralised and decentralised electrolysis options"

*Thanks for spotting this! Changed to: "Hydrogen production is modelled, distinguishing future investments in centralised and decentralised electrolysis options."*

13. Line 316 – 100% VRE share over the year? And is this the case even with an hourly resolution?

*Yes.*

14. Line 485 – regrading -> regarding

*Corrected, thanks!*

15. Line 544 – agricultural sector energy service demands don't change from 2018, how reasonable is this assumption?

*Thank you for pointing this out. The following has been added after "… are left unchanged from the 2018 level." to address the issue: "Since the sector is a large and export-led part of the Irish economy, these should be adjusted based on a specific scenario narrative."*

**Reviewer 2**

This paper provides a very welcome and well written example of what openness and transparency in modelling should look like. It clarifies how the model was developed, refers to an online and freely available repository of the model itself, goes in relative detail through the different sectors and parts of the model to highlight their structure and how they were built.

The model itself constitutes an integral "rebuild" of the existing TIMES model for the Republic of Ireland. It takes a new, flexible approach to the representation of both time-slices and regionality, and assesses the impact of deep penetration of VRE. It is a clear step forward and should offer robust and very much needed support in representing the government's new and ambitious targets.

In this context, the comments below should be considered minor and are aimed at further improving the high quality of the paper.

1. The space and importance given to the agricultural and land use sectors seems slightly on the light side. The authors note the importance of the sector for emissions in the country, as well as its complexity but do not describe it. While another paper is highlighted for the reader to refer to, additional detail and clarity on how the sector is treated & linked to other sectors in the model would be useful here too. Note, in addition, that some statements are confusing. The Abstract suggest that the authors / model will cover "transport, buildings, industry and agriculture" while those on the contents of the paper p3 line 69 describe the sections of the

paper as covering "supply, power, transport, residential, and industry"; and some sectoral paragraphs (e.g. supply) make statements about agricultural sector commodities.

*Thank you for pointing this out. We have expanded the description of the agricultural sector, as well as made previously confusing statements consistent.*

2. The authors suggest in that their approach considers the inclusion of equity in the model but do not explain this at any stage - while presumably linked to the residential sector "banding", this statement could be upheld in the sectoral discussions or otherwise clarified.

*We would like to clarify that we are not claiming that the model as it currently stands considers equity and distributional impacts of mitigation. However we do consider it an important priority for future model developments.*

3. On key feature of the model is its ability to consider flexibility between different levels of temporal and geographical detail. Considering the importance of this statement, the reader could expect to find additional information or a dedicated description of how this has been considered and how the authors ensure that energy balances, and transmission and distribution technologies are always aligned between versions.

*Thank you for pointing this out. We have expanded the corresponding section to include the relevant details.*

4. Consistency of demand drivers - an additional word explaining whether economic growth assumptions described in 2.5.2 are internally consistent would be useful. This also applies in other parts of the paper that discuss driver assumptions e.g. in relation to the housing demand estimates driven from Garcia Rodriguez (2020).

*We agree that internal consistency of the assumptions is important. The original section cited multiple sources without reflecting on this. We have clarified that that majority of these sources use the same input data contributing to the internal consistency of the assumptions.*

5. Supply sector - the authors state that attention to "best practice coding conventions" in section 3.1.2 but do not clarify the approach that is used, what it implies or what reference (if one exists) their approach is based on. A short additional paragraph would be useful here.

*The following paragraph has been added: "The model development team co-created a convention-dictionary workbook within the main repository that utilises automated three-letter coding conventions for technology, commodity and file name creation. This designed and followed code convention enables clean creation of process-sets, commodity-sets, results-sets and data automation routines for pre-processing and post-processing of data inputs, data outputs and results. Ultimately this makes the model architecture more transparent and easy to develop, enabling new users into the community of developers and peer review. Moreover, setting out predefined and intuitive commodity naming convention and a dictionary that is shared across TIM and LEAP-Ireland has multiple benefits for multi-model coupling, diagnostics and results reporting that is discussed."*

6. Bioenergy - future uses of bioenergy are gaining importance across increasing numbers of national and global modelling exercises. One key question that can raise significant difficulties is one of biomass sustainability. The current version of the paper / model seems to assume that biomass is carbon neutral. It would be useful to clarify this assumption, suggest why it is appropriate, clarify what definition of "biomass sustainability" is used (in broad terms at least) and consider if future versions of the model might usefully include the explicit representation of emissions linked to the use of different biomass commodities.

*We have added the following sentences to make the assumptions in the current model version explicit: "In the current version of TIM, biomass is assumed to be carbon neutral. However, future versions of the model may include different assumptions on emissions linked to use of biomass."*

7. Power sector - the authors suggest that the model is designed to take account of high levels of penetration of VRE, however the Power Sector description does not clarify what - if any - storage options are included in the model and how these are linked to VRE options to ensure system reliability. Could a paragraph to this effect be added?

*The following lines have been included: "The model technology database includes short-term storage battery technologies, medium-term pumped hydro storage, and long-term hydrogen storage. All these storage technology options are typically deployed in hourly unit-commitment capacity-expansion scenarios and provide ancillary services and balancing in 100% VRES scenarios."*

8. Transport - while detailed, the current description of the relationship between demand calculation, technology shares, changes of technology data over time vs. data considered constant over time (describing technology characteristics) is confusing and could be clarified. In particular, how the total demand is calculated and split across modes and what modal shifting is and is not possible, and under what conditions (i.e. in the core model or under particular scenarios), could be improved.

*Addressed these issues in multiple places in the transport sector description, including by adding the following statements:*

- *"Mode-specific relation between distance ranges is assumed constant throughout the modelling horizon…"*
- *"Total future passenger demand is obtained as a sum of mode-specific demand projections…"*
- *"Modal shift may be simulated within each travel distance band by defining scenario-specific shares for the different modes…"*

9. Residential - the BER rating system is stated to assume that living and non-living areas of buildings are heated to 21 or 18C respectively. Table 9 highlights how BER ratings C and below have significantly different internal temperature assumptions. It is not clear however how this links to modelling assumptions around lower technology or envelope efficiency. Is not clear if the lower grade buildings are expected to use the same amount of energy just delivering a poorer service than the higher grade ones? Or whether they instead also require higher levels of energy consumption to provide the lower temperatures?

*To readily compare different dwellings DEAP, the asset rating tool used to assign BERs in Ireland, assumes all living areas are heating to 21C and non-living are heated to 18C, however this overestimates the amount of residential heating according to the national energy balance in 2018 ( base year of model ). Also from previous Irish studies using measured data we know, thermally inefficient dwellings have colder internal temperatures than 21C/18C. Therefore, to better represent residential heating energy we have applied internal temperature assumptions (in Table 9) to different energy rated dwellings to reduce the national overestimation & better reflect measured data.*

*Despite lower temperatures in lower grade buildings, on average they still consume more energy then high grade buildings. This is not stated in the paper, as an average weighted calculation is used for existing dwellings i.e. the model currently does not disaggregate based on energy rating.*

*The technology performance & envelope efficiency is not dealt with directly in the model, instead we use the primary energy data from the BER database, however the BER primary energy data is based on fabric thermal efficiency and heating system efficiency. Therefore technology performance & envelope efficiency effect primary energy, which is the value used in the model.*

10. Section 4.2 states "Integration with key national data sources […] is a key strength of TIM". This suggests that the model can now be easily updated when new versions of key government publications are put forward. Is that the case? And if so, could this be stated more clearly?

*Changed to the following to avoid any confusion: "A key strength of TIM is that it draws from openly-available and regularly-updated national datasets, which allows for future updates, and alignment with other models drawing from the same sources."*

11. Please refer to the attached pdf with suggestions available in comments.

*Thank you. Addressed most as suggested.*